# Conformal Prediction for Dose-Response Models with Continuous Treatments

## Abstract

Understanding the dose-response relation between a continuous treatment and the outcome for an individual can greatly drive decision-making, particularly in areas like personalized drug dosing and personalized healthcare interventions. Point estimates are often insufficient in these high-risk environments, highlighting the need for uncertainty quantification to support informed decisions. Conformal prediction, a distribution-free and model-agnostic method for uncertainty quantification, has seen limited application in continuous treatments or dose-response models. To address this gap, we propose a novel methodology that frames the causal dose-response problem as a covariate shift, leveraging weighted conformal prediction. By incorporating propensity estimation, conformal predictive systems, and likelihood ratios, we present a practical solution for generating prediction intervals for dose-response models. Additionally, our method approximates local coverage for every treatment value by applying kernel functions as weights in weighted conformal prediction. Finally, we use a new synthetic benchmark dataset to demonstrate the significance of covariate shift assumptions in achieving robust prediction intervals for dose-response models.

## 1 Introduction

How can we determine the optimal dose for a patient to ensure the best therapeutic outcome? What is the impact of discounts in an online store on sales? What impact does $CO_2$ concentration have on local climates? At the core of each of these questions lies a shared causal idea: understanding the dose-response relation under continuous treatments to inform decision-making. In many cases, these decisions bear significant consequences, where relying solely on point estimates may be insufficient (Feuerriegel et al., 2024). Particularly in high-stakes situations, augmenting predictions with uncertainty quantification (UQ) can significantly improve decision-making processes (Feuerriegel et al., 2024). For instance, while the estimated causal effect of a continuous treatment may appear positive, prediction intervals could suggest a largely negative outcome for a specific individual. Such insights are crucial for deciding interventions. To tackle this, conformal prediction (CP) offers a robust solution for UQ, being both distribution-free and model-agnostic, with formal coverage guarantees (Vovk et al., 2022).

In this work, we seek to extend CP to UQ in dose-response models, aiming to aid decision-makers with more informed estimates to tackle such questions. We introduce a novel approach for deriving prediction intervals in the continuous treatment setting using weighted conformal prediction by combining propensity estimation with weighted conformal predictive systems. Furthermore, with the aid of a novel synthetic benchmark, we show how viewing the problem as a covariate shift approach provides coverage across all treatment values to help create more individualized dose-response curves.

## 2 Background

In this paper we expand upon the potential outcomes framework introduced in Rubin (2005), otherwise known as the Rubin framework to accommodate continuous treatments. Consider a continuous treatment variable $T \in [t_L, t_U]$ with a lower bound $t_L$ and upper bound $t_U$, observed covariates $X$, and potential outcomes $Y(t) \in \mathbb{R}$ representing the outcome that would be observed under treatment level $t$. The Conditional Average Dose-Response Function (CADRF) is defined as $\nu(x, t) =$

$E[Y(t)|X = x]$, the expected value over the Individual Dose-Response Functions (IDRF) for all individuals with observed $X$. Similar to Conditional Average Treatment Effects (CATE), to estimate the CADRF we make the following standard assumptions (Rubin, 2005; Hirano & Imbens, 2004):

- Unconfoundedness: $Y(t) \perp\!\!\!\perp T|X, \forall t \in T$. This assumption states that, conditional on the observed covariates, the treatment assignment is independent of the potential outcomes. In other words, there are no unobserved confounders that influence both the treatment assignment and the outcome.
- Overlap or positivity: $0 < P(T = t|X = x) < 1, \forall t \in T$ with $x \in X$. The overlap assumption ensures that for every covariate value $x$, there is a positive probability of receiving any treatment level. This is crucial for estimating treatment effects across the entire range of treatment levels.
- Consistency: $Y = Y(t)$ with probability 1. This assumption links the observed outcomes to the potential outcomes, stating that the observed outcome is equal to the potential outcome corresponding to the treatment received.

Quantifying the IDRF requires observing the $Y(t)$ for all possible treatment values. These treatment values are all counterfactuals and thus impossible to observe as we only can observe $Y$ for a single treatment value $t$ at a time. Furthermore for estimating the CADRF, likewise with CATE estimation, the distribution of the treatment assignment can bias the estimation (Hirano & Imbens, 2004). This distribution of the treatment assignment is called the propensity distribution, which was initially defined for binary treatments. Hirano & Imbens (2004) introduced the generalized propensity score (GPS) for continuous treatments that aims to unbias the CATE estimation for continuous treatments. The GPS is defined as $\pi(t_i|x) = f_{T|X}(T = t_i|X = x)$, which is the evaluation of $T = t_i$ on the conditional probability density function $T|X$ (Hirano & Imbens, 2004). If the treatment is independent of $X$, i.e. there are no confounders that influence treatment assignment, then $f_{T|X}$ is equal for all possible $X$. Furthermore, the treatment assignment is considered uniformly assigned between lower $t_L$ and upper $t_U$ possible treatment if $f_{T|X}$ represents the density function of the uniform distribution between $t_L$ and $t_U$. The GPS can then be used to mimic the randomly assigned treatment to estimate the unbiased CADRF (Wu et al., 2024).

The simplest method to estimate the CADRF is using an S-learner where a single learner is fit on both the covariates $X$ and the treatment $T$ to estimate $Y$. This approach provides a CADRF for each specific sample by keeping the covariates $X$ constant and changing $T$ to all different treatment values. However, if the treatment in the data is not uniformly assigned then the epistemic error can increase for specific treatment values $t_i$ and $X = x$ in low overlap regions or where $\pi(t_i|x)$ becomes very small. Consequently inferring $T = t_i$ in these regions would yield unreliable model estimates which should be communicated to ensure correct usage of a CADRF model.

The estimated $\widehat{IDRF}$ can also be seen as follows: $\widehat{IDRF} = \nu(x,t) + \epsilon_{a,IDRF}(x,t) + \epsilon_{e,IDRF}(x,t)$. The aleatoric uncertainty is symbolized by $\epsilon_{a,IDRF}(x,t)$ created by the inherent variability between individuals having the same covariates. $\epsilon_{e,IDRF}(x,t)$ symbolises the epistemic uncertainty coming from model specification and finite samples. Estimating both uncertainties creates the opportunity to estimate the ranges of the $\widehat{IDRF}$:

**Problem Definition** To accurately estimate the $\widehat{IDRF}$ for all possible treatment values we require correctly estimating both uncertainties for all treatment values equally, or more formally; for a specific significance level $\alpha$, lower treatment bound $t_L$, upper treatment bound $t_U$, and covariates $X$, we require prediction intervals $C(t, X)$ such that

$$\mathbb{P}(Y(t) \in C(X,t)) \geq 1 - \alpha, \quad \forall t \in [t_L, t_U] \tag{1}$$

This requirement necessitates prediction intervals that guarantee coverage for each possible treatment value individually.

## 3 RELATED WORK

Our proposed solution combines three different domains: propensity score methods, conformal prediction, and treatment effect or dose-response modelling.

**Propensity score methods**, introduced by Rosenbaum & Rubin (1983), have become widespread in causal inference, especially in observational studies. These methods aim to balance confounders across treatment groups, reducing bias in **treatment effect estimates**. Hirano & Imbens (2004) generalized this propensity score to continuous instead of binary treatments, introducing the generalized propensity score and building the foundation for causal inference with continuous exposures. Wu et al. (2024) used the generalized propensity score for matching continuous treatments to debias the treatment assignment and more accurately estimate the average dose-response curve for all treatment values. Other approaches adapt machine learning techniques to dose-response modelling. For instance, Athey et al. (2019) developed generalized random forests for heterogeneous treatment effect estimation, adaptable to continuous treatments.

To provide UQ, this work adapts **conformal prediction**. Conformal prediction is a model-agnostic method introduced by Vovk et al. (2022) that constructs prediction intervals with guaranteed finite-sample coverage under distribution-free assumptions. Conformal prediction uses conformity scores to assess uncertainty. Various improvements, such as the adaptive version by Romano et al. (2019), have increased the flexibility and applicability to even heteroscedastic settings. Additionally, Lei et al. (2018) and Papadopoulos et al. (2002) introduced split conformal prediction, significantly improving computational efficiency. For scenarios involving covariate or distribution shifts, Tibshirani et al. (2019) introduced weighted conformal prediction to ensure coverage under mismatched training and testing data distributions, with additional work by Gibbs & Candes (2021; 2024) and Barber et al. (2023). By reweighting the calibration samples similar to weighted conformal prediction, Guan (2023) introduced localized conformal prediction where the prediction intervals are determined by calibration samples localized around the test sample. Vovk et al. (2019) also introduced conformal predictive systems (CPS); an extension of full conformal prediction that allows extracting predictive distributions instead of prediction intervals. More recently, Jonkers et al. (2024a) combined previous concepts, introducing weighted conformal predictive systems to also account for covariate shifts.

In causal inference, conformal prediction has mainly been applied to binary treatments. For instance, Lei & Candès (2021) were among the first to apply conformal prediction to treatment effects estimation in randomized experiments and confounded or observational data. Jonkers et al. (2024b) and Alaa & Ahmad (2024) extended this approach to the potential outcomes framework, providing uncertainty to quantify individual treatment effects. However, the use of conformal prediction in continuous treatment settings remains largely unexplored. Schröder et al. (2024) proposed a conformal prediction framework for prediction intervals of treatment effects for continuous treatment interventions. However, their approach mainly covers single-treatment interventions and is computationally intensive, requiring optimization per confidence level, treatment, and sample where they provide prediction intervals for a single treatment value. For a more in-depth analysis of Schröder et al. (2024), see Appendix E.

Our goal is to achieve predictive coverage across the entire range of the treatment variable in estimating the dose-response curve. To our knowledge, no existing UQ methods offer conformal prediction guarantees for dose-response models with continuous treatments. To address this gap, we propose a novel methodology that seeks to provide this coverage by integrating weighted conformal prediction with propensity score weighting thereby guaranteeing coverage for any treatment value in continuous treatment dose-response models.

# 4 METHOD

## 4.1 INTRODUCTION TO CONFORMAL PREDICTION

Before delving into our proposed method, we provide a formal introduction to conformal prediction (Jonkers et al., 2024a; Tibshirani et al., 2019). Conformal prediction offers a powerful method for constructing prediction intervals with guaranteed finite-sample coverage under distribution-free assumptions (Vovk et al., 2022). The key insight of conformal prediction lies in its use of a nonconformity measure to quantify the degree to which a new observation differs from previously observed data.

Let us consider a regression problem with the training data being $n$ independent and identically distributed (i.i.d.) data pairs $Z_1 = (X_1, y_1), ..., Z_n = (X_n, y_n)$, where $X_i \in \mathbb{R}^d$ represents a vector of $d$ features and $y_i \in \mathbb{R}$ the corresponding label. Consider $Z_{n+1} = (X_{n+1}, y_{n+1})$ a new

exchangeable point being the test observation to evaluate and provide prediction intervals. Conformal prediction aims to construct a prediction interval $\hat{C}(X_{n+1})$ such that

$$\mathbb{P}\{y_{n+1} \in \hat{C}(X_{n+1})\} \geq 1 - \alpha \tag{2}$$

for a pre-specified significance level $\alpha \in (0, 1)$ where the probability is calculated over the points $Z_i, i = 1, ..., n$.

To achieve this, we first define a nonconformity measure $S((X, y), Z_{1:n})$ that quantifies how different the pair $(X, y)$ is from a multiset $Z_{1:n} = \{Z_1, ..., Z_n\}$ of data points. The lower the nonconformity measure, the more the pair conforms to the multiset $Z_{1:n}$. The most commonly used nonconformity measure is the absolute error $S((X, y), Z_{1:n}) = |y - \hat{\mu}(X)|$ with $\hat{\mu}$ an estimator fitted on $Z_{1:n}$.

Next, for each possible value $y \in \mathbb{R}$ that $y_{n+1}$ could be, we compute the nonconformity scores:

$$R_i^y := S((X_i, y_i), \{(X_1, y_1), ..., (X_{i-1}, y_{i-1}), (X_{i+1}, y_{i+1}), ..., (X_n, y_n), (X_{n+1}, y)\}), i = 1, ..., n \tag{3}$$

$$R_{n+1}^y := S((X_{n+1}, y), \{(X_1, y_1), ..., (X_n, y_n)\}) \tag{4}$$

Finally, we construct the prediction interval containing all $y$ where (Jonkers et al., 2024a)

$$\hat{C}(X_{n+1}) = \left\{ y \in \mathbb{R} : \frac{\#\{i = 1, ..., n+1 : R_i^y \geq R_{n+1}^y\}}{n+1} \geq 1 - \alpha \right\} \tag{5}$$

Tibshirani et al. (2019) presented conformal prediction slightly differently by using quantile functions instead, which will be more convenient for weighted conformal prediction later on. Tibshirani et al. (2019) defines the $1 - \alpha$ quantile function as follows, where $F_R(y)$ represents the distribution of nonconformity scores $R_i^y$ consisting of a sum of point masses $\delta_a$ with mass at $a$ where $R^y \sim F_R(y)$ (Tibshirani et al., 2019). $F_R(y)$ can then be used to calculate probabilities:

$$Quantile(1 - \alpha; F_R(y)) = inf\{R_i^y : \mathbb{P}\{R^y \leq R_i^y\} \geq 1 - \alpha\} \tag{6}$$

$$F_R(y) = \frac{1}{n+1} \sum_{i=1}^n \delta_{R_i^y} + \frac{1}{n+1} \delta_\infty \tag{7}$$

Finally, we construct the prediction interval containing all $y$ where

$$\hat{C}(X_{n+1}) = \{y \in \mathbb{R} : R_{n+1}^y \leq Quantile(1 - \alpha; F_R(y))\} \tag{8}$$

This procedure guarantees that $P(y_{n+1} \in \hat{C}(X_{n+1})) \geq 1 - \alpha$ for any exchangeable distribution of the data and any choice of nonconformity measure (Tibshirani et al., 2019).

### 4.1.1 INDUCTIVE CONFORMAL PREDICTION

The previously mentioned conformal prediction approach is computationally heavy as it requires fitting $n \cdot \#\{\mathbb{R}\} + 1$ estimators $\hat{\mu}$. Inductive or split conformal prediction (ICP), introduced by Papadopoulos et al. (2002), tackles this computation issue by splitting the training sequence $Z_{1:n} = \{Z_1, ..., Z_n\}$ into two sets: the proper training set $Z_{1:m} = \{Z_1, ..., Z_m\}$ and the calibration set $Z_{m+1:n} = \{Z_{m+1}, ..., Z_n\}$. A single regression model $\hat{\mu}$ is fit on the proper training set while the nonconformity scores (e.g., $R_i = |y_i - \hat{\mu}(X_i)|, i = m+1, ..., n$) are generated from the calibration set. These scores are sorted in descending order denoted as $R_1^*, ..., R_{n-m}^*$. Then, for a new sample with features $X_{n+1}$, a point prediction is made $\hat{y}_{n+1} = \hat{\mu}(X_{n+1})$. Finally, given a target coverage of $1 - \alpha$, the prediction interval becomes

$$\hat{C}(X_{n+1}) = [\hat{y}_{n+1} - R_s^*, \hat{y}_{n+1} + R_s^*] \tag{9}$$

where $s = \lfloor \alpha(n - m + 1) \rfloor$ represents the $1 - \alpha$ quantile of the ordered nonconformity set with size $n - m$ (Jonkers et al., 2024a).

### 4.1.2 WEIGHTED CONFORMAL PREDICTION

Evaluating and requiring coverage guarantees for the dose-response model at all possible treatment values changes the test distribution compared to the training distribution. In the training data, all treatment values are sampled according to their (conditional) training distribution, which can be determined by other variables in the case of confounding. However, every treatment value is possible in testing, and thus, every treatment sample can be sampled. This mimics sampling a new test sample with the treatment value from a uniform distribution, which can be vastly different from the treatment distribution in the training data. Standard conformal prediction only guarantees coverage if the joint distribution of the new sample $Z_{n+1}$ and $Z_{1:n}$ remains the same under permutations, which is called the exchangeability assumption (Vovk et al., 2022; Tibshirani et al., 2019). This issue is called covariate shift; The features $X_{n+1}$ come from a different distribution compared to $X_{1:n}$, while the relation between $X$ and $y$ remains the same. More formally: $X_i \sim P_X$, $i = 1, ..., n$ and $X_{n+1} \sim \tilde{P}_X$ where $\tilde{P}_X \neq P_X$ while $y_i \sim P_{Y|X}$, $i = 1, ..., n$.

Weighted conformal prediction provides a solution to tackle this issue (Tibshirani et al., 2019). However, their main assumption is that the likelihood ratio between the training $P_X$ and the test covariate distribution $\tilde{P}_X$ is known, defined as

$$w(x) = \frac{d\tilde{P}(x)}{dP(x)} \tag{10}$$

The rationale is that they reweight the distribution of nonconformity scores $F_R(y)$ to make the nonconformity scores more exchangeable with the test population by using the following weights in equation 7 (Tibshirani et al., 2019):

$$p_i^w(X_{n+1}) = \frac{w(X_i)}{\sum_{j=1}^n w(X_j) + w(X_{n+1})} \qquad p_{n+1}^w(X_{n+1}) = \frac{w(X_{n+1})}{\sum_{j=1}^n w(X_j) + w(X_{n+1})} \tag{11}$$

$$F_R(y) = \sum_{i=1}^n p_i^w(X_{n+1})\delta_{R_i^y} + p_{n+1}^w(X_{n+1})\delta_\infty \tag{12}$$

Consequently, these weights adjust the distribution of nonconformity scores to give more weight to nonconformity scores that are more likely in the test set and vice versa while in standard conformal prediction, every $R_i$ has equal weight. Also, note that the weights $p^w(x)$ are normalized, cancelling out any constant terms resulting in $w(x)$ being proportional to $w(x) \propto \frac{d\tilde{P}(x)}{dP(x)}$. An extension to split weighted conformal prediction can be done similarly as in section 4.1.1 (Tibshirani et al., 2019).

### 4.1.3 CONFORMAL PREDICTIVE SYSTEMS

In some cases, providing a prediction interval often does not suffice and a complete predictive distribution is required. The extension proposed by Vovk et al. (2019) produces a predictive distribution by arranging p-values, created using specific conformity measures, into a probability distribution function. A requirement to create a Conformal Predictive System (CPS) is to use a specific type of conformity measures [1] which include monotonic measures. Then, given the training data $Z_{1:n}$ and observed test sample $X_{n+1}$, we define an example of this specific conformity measure $S$ and conformity scores $R_i^y$ similar as in equations 3 and 4:

$$S((X, y), Z_{1:n}) = y - \hat{\mu}(X) \tag{13}$$

With $\hat{\mu}$ an estimator fitted on the training set $Z_{1:n}$. $R_i^y$ and $R_{n+1}^y$ are then similarly defined as in equation 3 for a CPS. Then, as defined in Vovk et al. (2022) we can define a predictive distribution Q for value $y$, using a distribution of nonconformity scores $F_R(y)$ of $y$ to calculate $\mathbb{P}$, similarly to the quantile function in equation 6 as follows:

$$Q_R(y, \phi) = \mathbb{P}_{F_R(y)}\{R^y < R_{n+1}^y\} + \phi \cdot \mathbb{P}_{F_R(y)}\{R^y = R_{n+1}^y\} \tag{14}$$

Where $\phi$ is a random number sampled from a uniform distribution between 0 and 1 to ensure a smooth predictive distribution. Using the same approach as section 4.1.2, these conformal predictive

---

[1] For the specific definition see Vovk et al. (2020)

systems can be expanded to weighted conformal predictive systems by adjusting $F_R(y)$ to account for the covariate shift (Jonkers et al., 2024a).

Additionally, conformal predictive systems also suffer from computational issues, therefore Vovk et al. (2020) introduced split conformal predictive systems to tackle the same issues in a way analogous to section 4.1.1.

### 4.2 PROPOSED METHODOLOGY: PROPENSITY WEIGHTED CONFORMAL PREDICTION

Taking into account the background knowledge of conformal prediction, we first need to formally define the target distribution to tackle our problem definition. A CADRF model $\hat{\nu}(X, T)$ is trained on triples $(X, T, Y)$ with X $d$-dimensional observed covariates $X \in \mathbb{R}^d \sim P_X$ and continuous treatment variables $T \in [t_L, t_U] \sim P_{T|X}$ to predict responses $Y \in \mathbb{R} \sim P_{Y|T,X}$. $P_X$ represents the covariate distribution, $P_{T|X}$ represents the observational conditional treatment distribution given confounders $X$, and $P_{Y|T,X}$ represents the outcome distribution. $P_{T|X} = P_T$ if there are no confounders for T. A CADRF model will be used to query the dose-response for all $T \in [t_L, t_U]$, creating an interventional distribution $\tilde{P}_T$. As every treatment value $t$ is equally likely in this query we can define $\tilde{P}_T = \tilde{P}_{T|X} = Uniform(t_L, t_U)$.

To attain marginal coverage across the interventional test set for a CADRF we can use weighted conformal prediction (Tibshirani et al., 2019). This requires defining the weights $w$ for $X_i$ and treatment value $t$ using equation 11, which we will call the global ($g$) propensity ($p$) weights $w_{g,p}$:

$$w_{g,p}(X_i, T_i) = \frac{d\tilde{P}_{X,T}(X_i, T_i)}{dP_{X,T}(X_i, T_i)} = \frac{d\tilde{P}_{T|X}(X_i, T_i)d\tilde{P}_X(X_i)}{dP_{T|X}(X_i, T_i)dP_X(X_i)} = \frac{d\tilde{P}_{T|X}(X_i, T_i)dP_X(X_i)}{dP_{T|X}(X_i, T_i)dP_X(X_i)}$$

$$= \frac{d\tilde{P}_{T|X}(X_i, T_i)}{dP_{T|X}(X_i, T_i)} = \frac{f_{U(t_L, t_U)}(T_i)}{\pi(T_i|X_i)} = \frac{\frac{\mathbb{1}_{[t_L, t_U]}(T_i)}{t_U - t_L}}{\pi(T_i|X_i)} \propto \frac{\mathbb{1}_{[t_L, t_U]}(T_i)}{\pi(T_i|X_i)} \tag{15}$$

with $\mathbb{1}_{[t_L, t_U]}(T_i)$ the indicator function for $T_i \in [t_L, t_U]$.

~~We~~ For simplicity, we assume that there is no distribution shift for $X$ and thus $\tilde{P}_X(X_i) = P_X(X_i)$ (The covariate shift approach for $X$ is detailed in Appendix D.1). Additionally, $f_{U(t_L, t_U)}$ is the probability density function for the uniform distribution. We also define the propensity function $\pi(T_i|X_i)$ as the probability density function for $P_{T|X}(T_i)$ as specified in Section 2. To generate the prediction intervals at treatment value $t$ for a new sample $X_{n+1}$ the weights change to $w_{g,p}(X_{n+1}, t) = \frac{1}{\pi(t|X_{n+1})}$. According to the weighted exchangeability defined in (Tibshirani et al., 2019), this guarantees marginal coverage over the interventional distribution, for all $T \in [t_L, t_U]$, and $X \sim P_X$. Tibshirani et al. (2019) also suggested a method to attain local coverage around a predetermined target point $x_0$ using weighted conformal prediction. Consequently, this can provide varying prediction intervals for different values of $x_0$ providing another heteroscedastic approach. The proposed weights, which we call the local ($l$) weights $w_l$, utilize kernel functions with bandwidth parameter $h$:

$$w_l^{x_0}(X_i) \propto K\left(\frac{X_i - x_0}{h}\right) \tag{16}$$

These weights then guarantee

$$\mathbb{P}_{x_0}\{Y_{n+1} \in \hat{C}(X_{n+1}; x_0)\} \geq 1 - \alpha \tag{17}$$

This assures coverage *around* $x_0$, but $x_0$ must be determined beforehand. Additionally, if a new $x_0$ must be evaluated, a new calibration procedure must be performed which should be considered when applying it to general regression use cases. However, for this work, the target interventional treatment distribution is known in advance and can all be computed before deployment. Consequently, for a target treatment value $t$ we can define $w_l^t(T_i) \propto K(\frac{T_i - t}{h})$ instead.

The local weights guarantee coverage where $d\tilde{P}_T(T_i)/dP_T(T_i) \propto K(\frac{T_i - t}{h})$. To adjust the local weights for a CADRF model we need to be aware of the covariate shift introduced by evaluating the interventional distribution and thus must combine $w_{g,p}$ with $w_{local}$ to achieve weighted exchangeability. These new weights are defined as $w_{l,p}$ for target treatment $t$:

$$w_{l,p}^t(X_i, T_i) \propto \frac{\mathbb{1}_{[t_L, t_U]}(T_i)K\left(\frac{T_i - t}{h}\right)}{\pi(T_i|X_i)} \tag{18}$$

To generate the prediction intervals for target treatment $t$ for a new sample $X_{n+1}$ the weights are then $w_{l,p}^t(X_{n+1}, t) = \frac{\mathbb{1}_{[t_L, t_U]}(T_i) K((t-t/h))}{\pi(t|X_i)} = \frac{\mathbb{1}_{[t_L, t_U]}(T_i)}{\pi(t|X_i)}$, which is equal to $w_{g,p}^t(X_{n+1}, t)$. By using these weights in a weighted conformal prediction framework, we provide a solution to the problem definition in Section 2. Theoretical coverage results of our approach are shown and discussed in Appendix A.

## 5  EXPERIMENTS

### 5.1  SYNTHETIC DATA

We evaluate the proposed approach on synthetic data as evaluating the true individual dose-response curve requires knowing the counterfactuals which is not feasible in real-world data.

We used three experimental setups using synthetic data, each having different scenarios that change specific parameters. Setup 1 is inspired by Wu et al. (2024) and Setup 2 follows the experimental setup of Schröder et al. (2024). Both Setup 1 and 2 are clarified in Appendix B. Setup 3 is novel, proposed by us, which mimics a situation where, for every scenario, two different possible dose-response functions are possible that each depends on the covariates, resulting in heavy confounding and thus limited overlap.

For each scenario (over the different setups), 5000 samples were generated using 50 different random seeds resulting in 50 datasets for each scenario. These datasets were split into 25% test (1250), 25% calibration (1250), and 50% training (2500) samples. For each scenario, two different $\alpha$ (significance values) were evaluated (i.e., 0.1 and 0.05 for a confidence of 90% and 95% resp.). Each sample in the test set is evaluated using 40 treatment values $t_0$ at equal intervals between the 2% and 98% training treatment value quantile to include varying treatment overlap regions and to mimic the uniform treatment sampling. In the results, the coverage of all treatment values and all samples in the test set are aggregated to a single mean coverage for each experiment, resulting in 50 mean coverage results for every method and scenario.

#### 5.1.1  SETUP 3

Setup 3 is a new experimental setup proposed in this work to underline the importance of compensating for confounding in UQ for CADRF. The covariates are independently sampled from a normal distribution. The treatment $T$ is confounded by two variables, determining the mean of the treatment assignment distribution:

$$X_1, X_2, X_3 \sim \text{Normal}(0, 5) \qquad T \sim \text{Normal}(X_2 + 0.1 \cdot X_1, 4)$$

The two scenarios have slightly different outcome distributions, as shown in Table 1. The idea is the same for both scenarios; The individual dose-response function is truly conditional and thus equal treatment values between different individuals or samples do not necessarily translate to each other. In total, there are four different possible dose-response functions depending on the covariates. Furthermore, there is heavy confounding resulting in limited samples where $T - X_2$ yields high values that in turn create large outcome values. This creates an opportunity for high epistemic uncertainty and limited overlap. For scenario two, the aleatoric uncertainty is also heteroscedastic based on $X_3$ forcing solutions to look beyond the treatment value to quantify uncertainty.

| Scenario | Outcome Distribution |
|:---:|:---:|
| 1 | $Y \sim sign(X_3) \cdot (2(T - X_2))^2 + 33T \cdot sign(X_1) + \text{Normal}(0, 2)$ |
| 2 | $Y \sim sign(X_3) \cdot (2(T - X_2))^2 + 33T \cdot sign(X_1)$ $+ \frac{(sign(X_3)+1)}{2} \cdot \text{Normal}(0, 30) + \text{Normal}(0, 2)$ |

Table 1: The outcome distributions for setup 3

## 5.2 IMPLEMENTATION

In the case of synthetic data, the true propensity distribution, also known as the oracle distribution, is available. However, in real-world applications, the true propensity distribution is mostly unknown. As a result, any method that relies on propensity is evaluated using both the oracle propensity distribution and an estimated propensity distribution in the experiments, denoted as "Oracle" and "Propensity" in the results respectively. The estimated distribution in this work is obtained using the Conformal Prediction System (CPS), though other propensity estimators could also be used. Do note that CPS quantifies total uncertainty and thus also includes the epistemic uncertainty while ideally only the aleatoric uncertainty is included. Additionally, this propensity distribution estimate is not completely guaranteed to be equal to the true conditional propensity distribution, which we theoretically need to get complete finite sample guarantees of validity. Although, in practice, this can still be a valid approximation. A learner is trained on the covariates $X$ to predict the treatment assignment $T$, deemed the propensity learner. Subsequently, a CPS is calibrated for this learner using the calibration set as it is more practical to extract an empirical density distribution compared to standard conformal prediction. Since CPS produces an empirical density distribution being a sum of Dirac delta distribution similar to $F_R$, kernel density estimation (KDE) is applied to derive a continuous propensity density function for a treatment value $t$, given covariates $X_i$. Do note that KDE interpolates the density and depending on the KDE parameters may introduce additional epistemic error, which is a drawback of estimating the propensity in this manner. The implementation and a computational discussion for Global and Local Propensity WCP is presented in Appendix C.1 and our propensity estimation in Appendix C.2.

For the evaluation, several baseline methods were tested and compared, including Gaussian Process, CatBoost with Uncertainty (Duan et al., 2019), Standard Conformal Prediction, and Locally Weighted Conformal Prediction (WCP Local, using weights $w_l$). For the proposed propensity methods we included both variations, using their respective weights: Global Propensity-Weighted Conformal Prediction (WCP Global Oracle and WCP Global Propensity, using $w_{g,p}$) and Local Propensity-Weighted Conformal Prediction (WCP Local Oracle and WCP Local Propensity, using $w_{l,p}$). The Gaussian Process was included in the comparison due to its widespread use for UQ in regression problems assuming a normal error distribution (Fiedler et al., 2021). All other approaches used a CatBoost model for the base CADRF learner, chosen for its strong out-of-the-box performance (Dorogush et al., 2018). As a result, the "CatBoost with Uncertainty" method was incorporated as a baseline for comparison of UQ.

The propensity learner employed in the propensity-weighted approaches was a `CatboostRegressor` with 4000 iterations and default hyperparameters. Similarly, the CADRF models were a CatBoost model with 5000 iterations and default hyperparameters. The CatBoost with Uncertainty approach used the same underlying CatBoost model as the other methods to ensure consistency. For the locally weighted conformal approaches, a Gaussian kernel (Theodoridis, 2015) was employed to represent local coverage. The bandwidth parameter for the kernel was set as $h = 2 \cdot (0.2 \cdot \sigma_{\hat{\pi}})^2$, where $\sigma_{\hat{\pi}}$ denotes the standard deviation of the estimated propensity distribution.

## 5.3 RESULTS

Figure 1 presents the coverage bar plots across all methods for Setup 3 Scenario 1 on the test set. More evaluations and CADRF RMSE on all setups and scenarios can be found in Appendix F. The bar plots in Figure 1 clearly illustrate the impact of covariate shift in the treatment on coverage guarantees for methods that did not account for this shift. All propensity-weighting methods assumed uniform treatment sampling during evaluation, mimicking the interpretation of a dose-response curve for decision-making for all treatment values, keeping their coverage guarantees.

As can be seen in Figure 1, the global propensity-weighting method shows a high variance in coverage across different experiments. This variance arises due to the calibration process, which considers all possible treatment values between $t_L$ and $t_U$, including those with minimal or no overlap. Depending on the calibration and test set split, certain samples may receive a significantly large likelihood ratio, thereby assigning considerable weight to those values according to Equation 12. This inflates the size

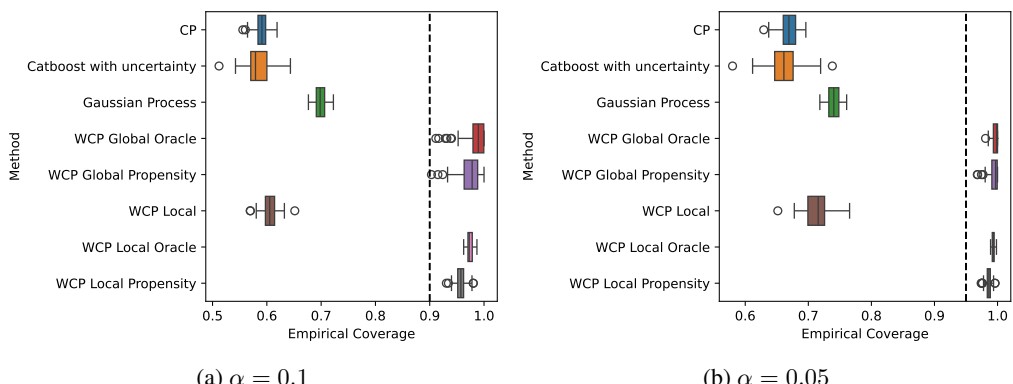

(a) $\alpha = 0.1$                                    (b) $\alpha = 0.05$

Figure 1: Barplot of the mean coverage calculated over 40 treatment values in 50 experiments for setup 3 scenario 1. Black dotted line is the ideal coverage.

of the prediction intervals, leading to conservative estimates. The oracle estimates are also notably more conservative, as they tend to provide narrower propensity distributions. This increases the frequency of large likelihood ratios when compared to the estimated propensity distribution, where the epistemic uncertainty of the propensity learner is also taken into account by the CPS procedure. On the contrary, for a new sample, the local propensity method uses calibration samples with treatment values close to the predefined value $t_0$ and weighting the propensities as well. Our presented approach uses more comparable calibration samples rather than the entire dataset, resulting in more conditional prediction intervals, provided there are enough calibration samples. Our method thus combines the strengths of both the local and the propensity weighting techniques. These trends are further supported in Figure 2, which shows the prediction intervals for all weighting methods alongside the treatment assignment distribution for a specific test observation. This example highlights the necessity of the uniform treatment sampling assumption for the evaluation of dose-response curves, as both the local weighting method and standard conformal prediction produce inaccurate prediction intervals in regions with low treatment overlap. In these regions, there is insufficient data to support predictions for the model, making these predictions unreliable. Consequently, propensity-weighted methods produce much larger prediction intervals in these areas to compensate for this lack of data support. If there is almost no support or extremely low propensity values, then the propensity-weighted methods provide intervals with an infinite width to show that there is no support in these regions. It is important to note, however, that these intervals may be overly conservative if the model has indeed generalized effectively in such regions. The only way to validate this is through additional data collection in these areas to confirm the model's performance.

Note that Schröder et al. (2024) also introduced a conformal prediction method to provide prediction intervals in the continuous treatment setting. However, we did not include a direct comparison in this study due to the high computational complexity of their approach, which would require several years to complete the same experiments we executed in a matter of hours. For a more detailed comparison, including a discussion of the difference in assumptions and methodologies, see Appendix E.

Implementing local propensity weighting in practice is less straightforward as it involves calibrating for a set of predefined treatment values and either storing these models for later use during inference or performing this action in parallel. This has the advantage that it allows conditional prediction intervals to be calculated more quickly during inference. However, a drawback is that evaluating a treatment value not included in the predefined set requires recalibration, and must be considered for inference. Still, this approach is particularly useful in fields like drug dosing, where treatment ranges are often predefined and personalized CADRF is highly relevant or where inference of new treatment values is not time-critical. Additionally, an important factor to consider is the effective sample size $\hat{n}$ in local propensity weighting (Tibshirani et al., 2019; Jonkers et al., 2024a). Reweighting $F_R(y)$ can significantly reduce the effective sample size, which increases variability in empirical coverage compared to standard conformal prediction. This issue is especially pronounced in regions with low treatment overlap, where the effective sample size can become extremely small. However, as prediction intervals with infinite length are possible using weighted conformal prediction, these infinite intervals additionally provide information to the user where the model cannot be trusted

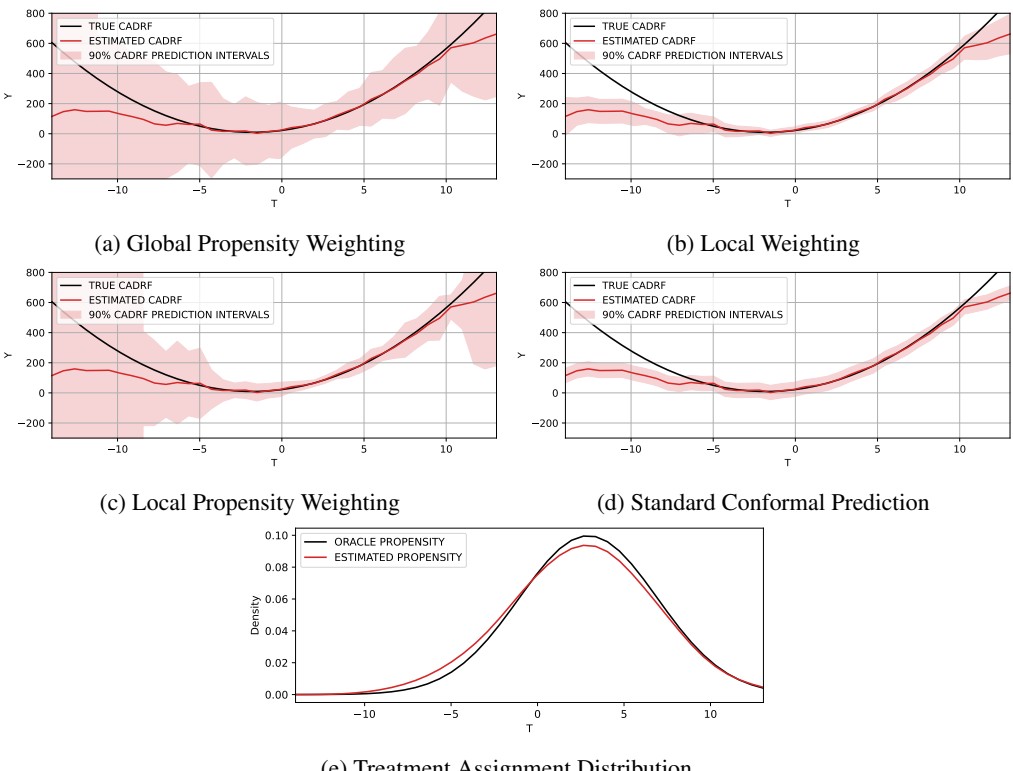

(a) Global Propensity Weighting

(b) Local Weighting

(c) Local Propensity Weighting

(d) Standard Conformal Prediction

(e) Treatment Assignment Distribution

Figure 2: CADRF UQ Example on Setup 3 Scenario 1 using estimated propensity

adding an interpretability layer to the UQ. In the current work, only an S-learner was used as a CADRF estimator which could influence the epistemic error, so in future work, more specialised dose-response models can be used to reduce the interval widths and provide even more informative prediction intervals.

Our current approach can be readily extended by incorporating other conformal prediction frameworks that support weighted conformal prediction, such as adaptive conformal prediction (Romano et al., 2019) or weighted conformal predictive systems (Jonkers et al., 2024a). Additionally, the weighting can be further expanded or changed to account for other types of covariate shifts in a similar manner or serve different purposes such as evaluating interventions of causal effects, thus broadening the applicability of the proposed method, as detailed in Appendix D.

## 6 CONCLUSION

In this work, we have introduced a novel approach to weighted conformal prediction for UQ in dose-response models, utilizing propensity estimation and kernel functions as weights for the likelihood ratio. Alongside a newly proposed synthetic dataset, our approach highlights the necessity of compensating for the covariate shift in the treatment assignment when evaluating dose-response models across all possible treatment values. This is achieved by assuming uniform treatment sampling during testing, similar to methods used in discrete treatment effect estimation. Additionally, by leveraging conformal predictive systems to estimate propensity distributions, we offer a practical solution to implement UQ in continuous dose-response estimation for various practical use cases.

Our contribution not only adds to the field of dose-response modelling but also facilitates delivering reliable, individualized dose-response functions. Our approach has the potential to aid decision-making for personalized dosing in fields such as marketing, policy-making, and healthcare. With this UQ for continuous treatments, we are one step closer to achieving truly personalized interventions that optimize outcomes for individuals.

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

## A  FINITE SAMPLE COVERAGE GUARANTEES

For counterfactual prediction intervals, the ideal goal is to achieve the following general conditional coverage guarantee:

$$\mathbb{P}_{Y \sim P_{Y|T=t, X=x}}(Y(t) \in \hat{C}(x,t)|X = x) \geq 1 - \alpha, \text{ where } t \in [t_L, t_U] \tag{19}$$

, which, under the *strong ignorability assumption*, is equivalent to:

$$\mathbb{P}_{Y \sim P_{Y|T=t, X=x}}(Y \in \hat{C}(x,t)|X = x, T = t) \geq 1 - \alpha. \tag{20}$$

However, constructing non-trivial prediction intervals with such conditional guarantees is generally impossible without additional modeling assumptions, as shown in Foygel Barber et al. (2021). Even under the relaxed conditional guarantee, where conditioning is only on the treatment value, as in binary treatment settings (Lei & Candès, 2021):

$$\mathbb{P}_{Y \sim P_X \times P_{Y|T=t, X}}(Y \in \hat{C}(X,t)|T = t) \geq 1 - \alpha, \tag{21}$$

the problem persists when the treatment variable $t$ is continuous.

### A.1  PROPOSED FRAMEWORK

To address this challenge, we introduce a distribution shift in the treatment variable by moving from the generalized propensity distribution to a user-specified interventional distribution, $T_{n+1} \sim \tilde{P}_{T|X}$. We then leverage the weighted conformal prediction (WCP) framework to construct prediction intervals. This approach allows us to build on prior theoretical coverage results under both oracle and estimated likelihood functions(Tibshirani et al., 2019; Lei & Candès, 2021).

Table 2 outlines the two interventional distributions utilized in this work: global propensity, local propensity, and $\delta$-propensity (Dirac delta). The latter corresponds to a hard intervention. Relaxing the $\delta$-propensity to the local propensity enables the construction of non-trivial prediction intervals (see Remark4). Notably, when $T \in \{0, 1\}$, our approach under $\delta$-propensity aligns with the counterfactual inference framework for binary treatments proposed in Lei & Candès (2021).

Table 2: Translation of general interventional distribution framework to WCP global, local, and $\delta$-propensity.

| General | Global propensity | Local propensity | $\delta$-propensity |
|---|---|---|---|
| $\tilde{P}_{T|X}$ | $Uniform(t_L, t_U)$ | $\dfrac{\mathbb{1}_{[t_L, t_U]}(T) K\left(\frac{T-t}{h}\right)}{\int_{t_L}^{t_U} \mathbb{1}_{[t_L, t_U]}(T) K\left(\frac{T-t}{h}\right) dT}$ | $\delta(T - t)$ |
| $w(X, T)$ | $\dfrac{\mathbb{1}_{[t_L, t_U]}(T)}{\pi(T|X)}$ | $\dfrac{\mathbb{1}_{[t_L, t_U]}(T) K\left(\frac{T-t}{h}\right)}{\pi(T|X)}$ | $\dfrac{\delta(T-t)}{\pi(T|X)}$ |
| $\hat{w}(X, T)$ | $\dfrac{\mathbb{1}_{[t_L, t_U]}(T)}{\hat{\pi}(T|X)}$ | $\dfrac{\mathbb{1}_{[t_L, t_U]}(T) K\left(\frac{T-t}{h}\right)}{\hat{\pi}(T|X)}$ | $\dfrac{\delta(T-t)}{\hat{\pi}(T|X)}$ |

## A.2 PROPOSTION: FINITE-SAMPLE GUARANTEES

**Proposition 1** (following Tibshirani et al. (2019); Lei & Candès (2021)). *Assume $(X_i, T_i, Y_i) \overset{i.i.d.}{\sim} P_X \times P_{T|X} \times P_{Y|T,X}$, $i = 1, ..., n$; the likelihood ratio $w(X,T) \propto \frac{d\tilde{P}_{T|X}}{dP_{T|X}}$; and the estimated likelihood ratio $\hat{w}(X,T)$. Using WCP to construct $\hat{C}(X,T)$, the following finite-sample bounds apply:*

**S1.** **(Oracle Likelihood Ratio)** *If $\hat{w}(\cdot,\cdot) = w(\cdot,\cdot)$, i.e. oracle likelihood ratio function; then,*

$$1 - \alpha \leq \mathbb{P}_{(X,T,Y) \sim P_X \times \tilde{P}_{T|X} \times P_{Y|T,X}} \{Y \in \hat{C}(X,T)\} \tag{22}$$

**S2.** **(Finite Sample with Regularity Conditions)** *If $\hat{w}(\cdot,\cdot) = w(\cdot,\cdot)$; the non-conformity scores $S_i$ have no ties almost surely; $\tilde{P}_{T|X} \times P_X$ is absolutely continuous with respect to $P_{T|X} \times P_X$; and $(\mathbb{E}_{(X,T) \sim P_X \times P_{T|X}}[w(X,T)^r])^{\frac{1}{r}} \leq M_r < \infty$ where $r > 0$ and $M_r$ denotes the upper bound of the $r$-th moment of the likelihood ratio; then,*

$$1 - \alpha \leq \mathbb{P}_{(X,T,Y) \sim P_X \times \tilde{P}_{T|X} \times P_{Y|T,X}} \{Y \in \hat{C}(X,T)\} \leq 1 - \alpha + c n^{\frac{1}{r-1}} \tag{23}$$

*where $c$ is an arbitrary positive constant depending on $M_r$ and $r$.*

**S3.** **(Estimated Likelihood Ratio)** *If $\hat{w}(\cdot,\cdot) \neq w(\cdot,\cdot)$; $\Delta_w = \frac{1}{2}\mathbb{E}_{(X,T) \sim P_X \times P_{T|X}}[|\hat{w}(X,T) - w(X,T)|]$; $(\mathbb{E}_{(X,T) \sim P_X \times P_{T|X}}[\hat{w}(X,T)^r])^{\frac{1}{r}} \leq M_r < \infty$; and further assuming the same assumptions as in S2.; then,*

$$1 - \alpha - \Delta_w \leq \mathbb{P}_{(X,T,Y) \sim P_X \times \tilde{P}_{T|X} \times P_{Y|T,X}} \{Y \in \hat{C}(X,T)\} \leq 1 - \alpha + \Delta_w + c n^{\frac{1}{r-1}} \tag{24}$$

*Proof.* We can reformulate our problem as a covariate shift scenario by treating the treatment variable as part of the covariates, i.e., defining $X^* = [X, T]$. Under this transformation:

- The proof for setting **S.1** follows directly from Theorem 2 in Tibshirani et al. (2019).

- The proof for setting **S.2** aligns with Proposition 1 in Lei & Candès (2021). While their work focuses explicitly on split-weighted conformalized quantile regression (CQR) (Romano et al., 2019), the argument extends to WCP because it only depends on the weighted exchangeability of nonconformity scores and the boundedness of the likelihood ratio function.

- Similarly, the proof for setting **S.3** follows from Theorem 3 in Lei & Candès (2021), along with its corresponding derivation.

□

**Remark 1.** *$r$ specifies which moment of the likelihood ratio $w(X,T)$ is being considered. Larger $r$ corresponds to stricter regularity conditions on $w(X,T)$. $M_r$ defines the upper bound on the $r$-th moment of $w(X,T)$, ensuring the likelihood ratio does not grow too large and remains well-behaved.*

**Remark 2.** *Note that the term $c n^{\frac{1}{r-1}}$, represents the upper bound of the expectation of maximum weight (probability), i.e., $\mathbb{E}\left[\max_{i \in [1,...,n] \cup \{\infty\}} p_i^w(X_{n+1})\right]$, which under no covariate shift is equal to $\frac{1}{n+1}$ the upper bound of unweighted conformal prediction.*

**Remark 3.** *The bounding condition assumed in S.2 and S.3 in Proposition 1, $(\mathbb{E}[w(X,T)^r])^{\frac{1}{r}} \leq M_r < \infty$, that $\mathbb{E}[w(X,T)^r] < \infty$ implies that $\mathbb{P}_{(X,T) \sim P_X \times P_{T|X}}(w(X) < \infty) = 1$ and $\mathbb{E}[w(X)] < \infty$ (Lei & Candès, 2021), i.e. $P_X \times \tilde{P}_{T|X}$ is absolutely continuous with respect to $P_X \times P_{T|X}$.*

**Remark 4.** *For setting **S.1**, the overlap or positivity assumption can be violated, i.e., $\frac{d\tilde{P}_{T|X}}{dP_{T|X}} = \infty$ in terms of the interventional distribution. However, this results in the trivial interval $(-\infty, \infty)$, since $w(X_i) = 0, \forall i \in [1, ..., n]$ and $w(X_{n+1}) = \infty$ resulting in $p_i^w(X_{n+1}) = 0, \forall i \in [1, ..., n]$ and $p_{n+1}^w = 1$.*

**Remark 5.** *Since inductive (or split) conformal prediction is a special case of conformal prediction, Proposition 1 also applies to inductive conformal prediction, which we use in our experiments.*

**Remark 6.** *With an estimated likelihood ratio under weighted CQR, our approach also follows the asymptotic double robustness result (see Theorem 1 (Lei & Candès, 2021)).*

# B SYNTHETIC DATA

## B.1 SETUP 1

For setup 1, inspired by Wu et al. (2024), six independent covariates are sampled from various distributions representing both continuous and discrete values:

$$X_1, X_2, X_3, X_4 \sim \text{Normal}(0, 1)$$
$$X_5 \sim \text{Uniform}[-2, 2] \text{ (Integer)}$$
$$X_6 \sim \text{Uniform}(-3, 3)$$

The treatment value is confounded by all variables in this setup and thus determined by a treatment function $T_\mu$. All scenarios share the same treatment function except for scenario 3, where a quadratic term was added. The treatment functions are shown in Table 3.

| Scenario | Treatment function |
|---|---|
| $1, 2, 4, 5, 6, 7, 8$ | $T_\mu = -0.8 + X_1 + 0.1X_2 - 0.1X_3 + 0.2X_4 + 0.1X_5 + 0.1X_6$ |
| $3$ | $T_\mu = -0.8 + X_1 + 0.1X_2 - 0.1X_3 + 0.2X_4 + 0.1X_5 + 0.1X_6 + \frac{3}{2}X_3^2$ |

Table 3: The treatment functions for all scenarios in setup 1.

The true assigned treatment value $T$ is then sampled from a treatment assignment distribution to add randomness and ensure some overlap in the simulated data. This treatment assignment distribution is different for various scenarios to evaluate the differences in the assumed distributions. The various functions are shown in Table 4

| Scenario | Treatment $T$ | Treatment Assignment Distribution |
|---|---|---|
| 1 | $9T_\mu + 17$ | Normal$(0, 5)$ |
| 2 | $15T_\mu + 22$ | StudentT$(df = 2)$ |
| 3 | $9T_\mu + 15$ | Normal$(0, 5)$ |
| 4 | $49\frac{e^{T_\mu}}{1+e^{T_\mu}} - 6$ | Normal$(0, 5)$ |
| 5 | $42\frac{1}{1+e^{T_\mu}} + 18$ | Normal$(0, 5)$ |
| 6 | $7log(|T_\mu| + 0.001) + 13$ | Normal$(0, 4)$ |
| 7 | $7T_\mu + 16$ | Normal$(0, 1)$ |
| 8 | $7T_\mu + 16$ | $20 \cdot$ Beta$(\alpha = 2, \beta = 8)$ |

Table 4: The propensity functions per scenario for Setup 1

Now, given both the covariates $X$ and the assigned treatment $T$ the outcome function is defined as a random variable sampled from a normal distribution with a variance of 5, with the mean a function

dependent on both the treatment and the covariates:

$$Y \sim -1 - (2X_1 + 2X_2 + 3X_3^3 - 20X_4 - 2X_5 + 20X_6)$$
$$- 0.1T(1 - X_1 + X_4 + X_5 + X_3^2) + 0.13^2|T|^3 sin(X_4) + \text{Normal}(0, 5)$$

### B.2 SETUP 2

Setup 2 tests the different treatment assignment distributions in the two different scenarios, which is the same experimental setup as proposed by Schröder et al. (2024). The covariates are sampled from a discrete uniform distribution. The treatment is sampled from the treatment assignment distributions shown in Table 5. The outcome function is sampled from a normal distribution with a mean determined by a sinus function based on both $X$ and $T$:

$$X \sim \text{Uniform}[1, 4] \text{ (Integer)}$$
$$Y \sim sin\left((0.05\pi)(T - X)\right) + \text{Normal}(0, 0.1)$$

| Scenario | Treatment Assignment Distribution |
|---|---|
| 1 | $T \sim p \cdot \text{Uniform}(0, 5X) + (1 - p)\text{Uniform}(5X, 40), p \sim \text{Bernoulli}(0.3)$ |
| 2 | $T \sim \text{Normal}(5X, 10)$ |

Table 5: The propensity functions per scenario for Setup 2

## C PROPENSITY DISTRIBUTION ESTIMATION ALGORITHM PSEUDOCODE AND COMPUTATIONAL ANALYSIS

### C.1 PROPENSITY-BASED WEIGHTED CONFORMAL PREDICTION PSEUDOCODE

Algorithm 1 presents the fit procedure for both the Local and the Global Propensity WCP, using their respective weights $w_{l,p}^t$ and $w_{g,p}^t$ for an array of treatment values we want to evaluate $t_{eval}$. The pseudocode is written for any Kernel, although in the experiments, we used the Gaussian kernel as presented in the methodology section. The pseudocode assumes either a pre-fitted propensity estimator $\hat{\pi}$ or having access to an Oracle estimator. The method used to fit the propensity estimator in this paper is presented in Appendix C.2. Algorithm 2 then presents how the prediction intervals for a significance level $\alpha$ are generated using both Local and Global Propensity WCP as the implementation is the same for both methods. The get_interval function is the prediction interval function of the WCP method.

**Algorithm 1** Fit and calibrate Local or Global Propensity WCP

1: **Input:** Training covariates $X_{tr}$, calibration covariates $X_{cal}$, training outcome $y_{tr}$, calibration outcome $y_{cal}$, training treatment values $T_{tr}$, calibration treatment values $T_{cal}$, calibrated PropensityEstimator or oracle $\hat{\pi}$, to evaluate treatments in array $t_{eval}$, kernel $K$, CADRF learner $\hat{\mu}$
2: Fit CADRF $\hat{\mu}$ on $(X_{tr}, T_{tr})$ to predict $y_{tr}$
3: Calculate propensities $\pi_{cal} = \hat{\pi}(X_{cal})$
4: **if** Global Propensity WCP **then**
5:     Calculate weights: $w_{g,p} = 1/\pi_{cal}$
6:     Define $WCP$ as Weighted Conformal Prediction with learner $\hat{\mu}$ and weights $w_{g,p}$ on $(X_{cal}, T_{cal}, y_{cal})$
7:     Calibrate $WCP$
8: **else if** Local Propensity WCP **then**
9:     **for** $t$ **in** $t_{eval}$ **do**
10:         Calculate weights: $w_{l,p}^t = K(T_{cal}, t)/\pi_{cal}$
11:         Define $WCP_t$ as Weighted Conformal Prediction with learner $\hat{\mu}$ and weights $w_{l,p}^t$ on $(X_{cal}, T_{cal}, y_{cal})$
12:         Calibrate $WCP_t$
13:     **end for**
14: **end if**
15: **Output:** Calibrated models $\{WCP_t : t \in t_{\text{eval}}\}$ for Local Propensity WCP or $WCP$ for Global Propensity WCP

---

**Algorithm 2** Provide uncertainty estimates Local and Global Propensity WCP

1: **Input:** Test sample $X_{n+1}$, calibrated PropensityEstimator or oracle $\hat{\pi}$, $k$ to evaluate treatments in array $t_{eval}$, kernel $K$, CADRF learner $\hat{\mu}$, calibrated $WCP_t$ for all $t$ in $t_{eval}$, significance $\alpha$
2: Calculate $\pi_{n+1} = \hat{\pi}(X_{n+1})$
3: Calculate weights $w = 1/\pi_{cal}$
4: **for** $t$ **in** $t_{eval}$ **do**
5:     Predict outcome: $\hat{\mu}(X_{n+1}, t)$
6:     Obtain prediction interval: $\hat{C}_{n+1}^t = \text{get\_interval}(WCP_t, (X_{n+1}, t), \alpha, w^t)$
7: **end for**
8: **Output:** Prediction intervals $\left[\hat{C}_{n+1,\alpha}^{t_{\text{eval},1}}, \ldots, \hat{C}_{n+1,\alpha}^{t_{\text{eval},k}}\right]$

---

## C.2 PROPENSITY DISTRIBUTION ESTIMATION PSEUDOCODE

Algorithm 3 presents the propensity distribution estimation using Conformal Predictive Systems (CPS). This results in a propensity distribution array $\pi_{arr}$ with the calculated propensity density for each sample in $X_{cal}$. $exp$ is the exponential function and $len(X)$ denotes the length of the array $X$.

---

**Algorithm 3** Estimating the Propensity Distribution

1: **Input:** ~~training~~ Training covariates $X_{tr}$, calibration covariates $X_{cal}$, training treatment values $T_{tr}$, calibration treatment values $T_{cal}$, Kernel Density Estimator $KD$
2: ~~fit~~ Fit propensity learner on $X_{tr}$ to predict $T_{tr}$
3: ~~calibrate CPS on~~ Calibrate CPS using $X_{cal}$ and $T_{cal}$
4: ~~Define~~ Initialize $\pi_{arr}$ ~~with~~ as an array of length $len(X_{cal})$
5: **for** $i = 1$ **to** $len(X_{cal})$ **do**
6:     ~~fit KD(~~Fit $KD$ on CPS($X_{cal,i}$~~)~~)
7:     Set $\pi_{arr}[i] = exp(KD(T_{cal,i}))$
8: **end for**
9: ~~return $\pi_{arr}$~~ **Output:** Propensity array $\pi_{arr}$

---

## C.3 Computational Overhead

The computational overhead is greatest for Local Propensity WCP due to the evaluation over multiple treatment values, so we will focus on this version. Let $m$ denote the number of treatment values in the evaluation array $t_{\text{eval}}$. In this case, the computational overhead compared to standard weighted conformal prediction (WCP) scales linearly with the number of treatment values, i.e., $O(m \cdot \text{WCP})$, where WCP refers to the cost of standard weighted conformal prediction. In addition, calculating the propensities $\pi_{\text{cal}}$ on the calibration set incurs an additional computational cost, which depends on the size of the calibration set and the chosen propensity estimator. This step can be done once beforehand, so it does not need to be repeated during each evaluation.

If the treatment values in $t_{\text{eval}}$ are known and fixed, the calibration for each treatment value can be precomputed and stored, resulting in saved $WCP_t$ models. This means that, during inference, the computational overhead is reduced to calculating the propensity for a single new sample once and performing $m$ predictions using the CADRF, followed by retrieving the prediction intervals for each treatment value using the pre-calibrated $WCP_t$. Thus, the inference overhead is $O(m)$ for a single inference, consisting of a propensity calculation and $m$ predictions and interval retrievals. In the case of a non-static or on-demand $t_{\text{eval}}$, the overhead is additive as we need $O(mWCP)$ calibrations and directly afterward $O(m)$ for the inference.

If there is no Oracle propensity estimator, we need to fit the propensity estimator, which, in our case, also involves fitting the Kernel Density Estimator (KDE) for each sample in $X_{\text{cal}}$, as detailed in Algorithm 3. This introduces an extra layer of computational overhead, which depends on the size of the calibration set and the output of the CPS, which is an empirical distribution of the treatment values for $x_{cal}$. The KDE fitting step needs to be performed for each element of $X_{\text{cal}}$, resulting in a complexity of $O(\text{len}(X_{\text{cal}}) \cdot \text{KDE})$, where $\text{len}(X_{\text{cal}})$ is the number of calibration samples and KDE denotes the cost of fitting the KDE.

## D Extensions and Applications of weighted conformal dose-response curves

Here, we discuss possible extensions and how the proposed method can be applied to various applications.

### D.1 Extensions

The paper's current setup assumes no covariate shift in the features $X$ between the training, calibration, and test set, i.e., $P_X = \tilde{P}_X$, to simplify the derivation of the propensity-based weights. However, in real-world applications, covariate shifts are much more common and can hamper the coverage guarantee of conformal prediction, and also thus our proposed method (Tibshirani et al. (2019)). If we assume $P_X \neq \tilde{P}_X$ in equation 15, we observe that this results in adding a multiplicative term that represents the likelihood term for the covariate shift in $X$. As such, both $w_{l,p}^t$ and $w_{g,p}$ can be easily adjusted to cover a covariate shift in the test set if the covariate shift is known or can be calculated, analogous to Tibshirani et al. (2019), resulting in the following new weights:

$$w_{g,p}(X_i, T_i) \propto \frac{\mathbb{1}_{[t_L,t_U]}(T_i)}{\pi(T_i|X_i)} \frac{d\tilde{P}_X}{d\tilde{P}_X} \tag{25}$$

and

$$w_{l,p}^t(X_i, T_i) \propto \frac{\mathbb{1}_{[t_L,t_U]}(T_i) K\left(\frac{T_i-t}{h}\right)}{\pi(T_i|X_i)} \frac{d\tilde{P}_X}{d\tilde{P}_X} \tag{26}$$

Furthermore, because the method is built using conformal prediction, the whole approach is model-agnostic. As such any possible CADRF model that provides a dose-response curve given features and treatment can be used and thus is not limited to the presented CADRF approach in this paper.

## D.2 APPLICATIONS

The classic application is in drug dosing, where the goal is to construct a dose-response curve for every individual to facilitate decision-making when determining an optimal dose for a new patient. In a clinical trial, especially phase 1 and phase 2 where the optimal dose is being determined, the weighted conformal dose-response curve can also act as a tool to analyse the results individually while having an estimate of the uncertainty estimates that is not biased by the treatment assignment distribution. It quantifies uncertainty for individual predictions, compensating for any treatment distribution bias. Furthermore, it highlights areas with insufficient data support with infinite prediction intervals, guiding decisions about whether further trials or treatments are necessary for specific patient subgroups. In the regions where there is support, the model predictions provide the CADRF estimate for this patient and the uncertainty regions show how the outcome would vary.

Treatment is not limited to healthcare. Treatment can be generalized as any intervention or action which opens applications in other domains. For example, in predictive maintenance, the model can optimize decisions by estimating the effect of operating pressure on the remaining useful life of equipment like valves. Similarly, in sales, it can help determine the ideal discount for specific clients to maximize the sold units, demonstrating flexibility in various domains.

## D.3 EXPLAINABILITY

The application potential is also not limited to actual treatments and interventions. The method can also be used for the explainability of a model. Suppose we fitted a regression model, regressing $X = [X_1, ..., X_m]$ on $Y$. $X$ is observed data; thus, any feature can be confounded or biased. By considering a feature $X_i$ as a treatment to "intervene" in a model, this method then provides uncertainty quantification on a Ceteris Paribus curve of a model in a similar manner to a dose-response curve [2]. This curve can then give unbiased uncertainty estimates of the "true" outcome for an individual sample if that sample would have had other values for this particular feature.

An example is shown in Figure 3 using Local Propensity WCP. This example is generated using the Boston Housing data available native in sklearn (Pedregosa et al., 2011), split into a training and calibration set using a 75/25 split. A CatBoostRegressor using 300 iterations is fitted on a training set, and a propensity CatBoostRegressor with the same number of iterations is fitted on the training set. A CPS is used and calibrated on the calibration set for the propensity distribution estimate, similar to the experimental setup in this paper. No hyperparameter tuning is applied for simplicity, so note that the epistemic uncertainty could be further reduced. The chosen feature for generating a ceteris paribus curve is MedInc, the median income, an important variable in predicting the median house value in this dataset. The figure is for a single data sample where all other variables of this sample are kept constant except for our "treatment" MedInc. In Figure 3, it is apparent that the prediction intervals go to infinity for MedInc values below 1 and above 6.5. This indicates that there is insufficient overlap to evaluate this sample for these values of MedInc, clearly showing a bias in the data distribution of MedInc, given the other features. Consequently, the predictions for a sample with these features but with a MedInc of, e.g., 8 cannot be trusted as the model is simply doing an interpolation in an out-of-bounds region. In the regions with support, i.e., around $1.5 < MedInc < 6.5$, we see that the model shows a linear relation with the median house value with relatively small uncertainty bounds. This analysis can be done for any other regression model in a likewise manner.

## E COMPARISON TO SCHRÖDER ET AL.

In comparison to the work of Schröder et al. (2024), our approach differs in several key aspects. First, the aim of their work is different from ours. The aim of Schröder et al. (2024) is to provide prediction intervals for the causal effect of treatment interventions where the treatment value is continuous. In our work, the goal is to provide prediction intervals for dose-response models instead of treatment interventions, answering a different causal question. However, adjusting our work to interventions

---

[2] A Ceteris Paribus curve visualizes a model's predictions while keeping all features constant except for one explanatory variable. The x-axis represents the explanatory variable, and the y-axis shows the corresponding predictions.

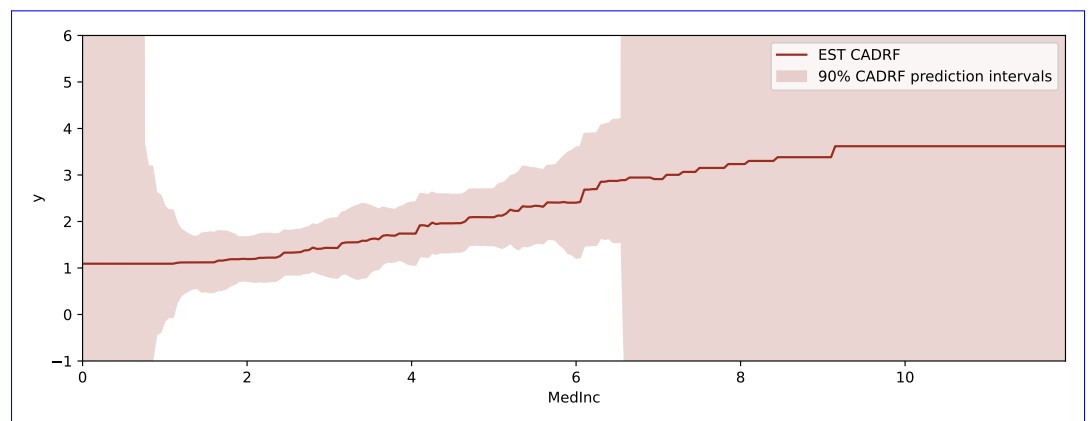

Figure 3: A Ceteris Paribus curve generated with Local Propensity WCP.

is possible; In the case of soft interventions, the target distribution propensity changes and thus substituting the current uniform distribution in the weights $w(x)$ with the new target propensity distribution covers the soft intervention case. For hard interventions, this is an evaluation for a single treatment value which is similar to the local propensity method, but for only that target treatment value. Secondly, their approach differs in their conformal prediction approach where they want to provide correct prediction intervals for a single sample, single $\alpha$ value, and single treatment using a mathematical solver based on the proposed weighted conformal prediction by Gibbs & Candes (2021). Thirdly, they frame the propensity or covariate shift differently as either a Dirac distribution for a hard intervention, or a different propensity distribution in the case of a soft intervention. This is a direct consequence of their aim to quantify the causal effect of a single intervention, compared to providing a dose-response model in our case which requires a uniform assumption. Fourthly, the experimental setup of Schröder et al. (2024) does not address the impact of a treatment covariate shift as shown by Figure 5 and Figure 6 where even standard conformal prediction (CP) achieves the required empirical coverage. Lastly, we also approach the propensity estimation in cases with unknown propensity as an uncertainty quantification problem and tackle it with conformal predictive systems. In the end, our approach offers a different solution on continuous treatment effects through dose-response modelling.

# F   ADDITIONAL RESULTS

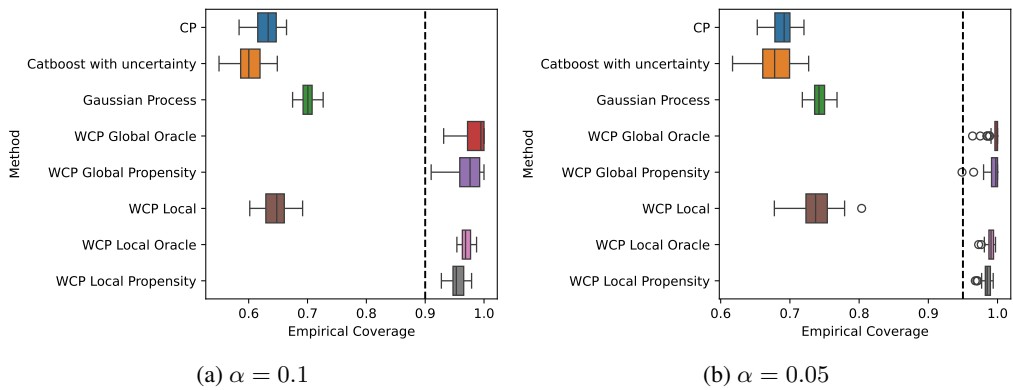

(a) $\alpha = 0.1$          (b) $\alpha = 0.05$

Figure 4: Barplot of the mean coverage calculated over 40 treatment values in 50 experiments for setup 3 scenario 2. Black dotted line is the ideal coverage.

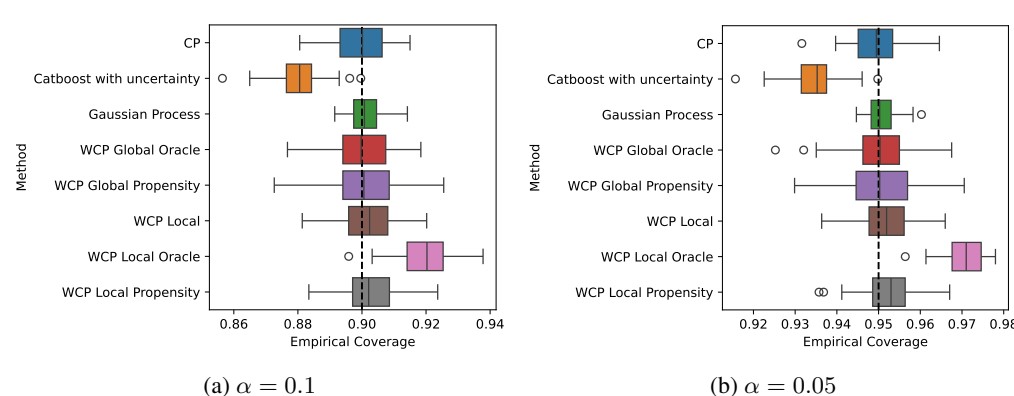

(a) $\alpha = 0.1$        (b) $\alpha = 0.05$

Figure 5: Barplot of the mean coverage calculated over 40 treatment values in 50 experiments for setup 2 scenario 1. Black dotted line is the ideal coverage.

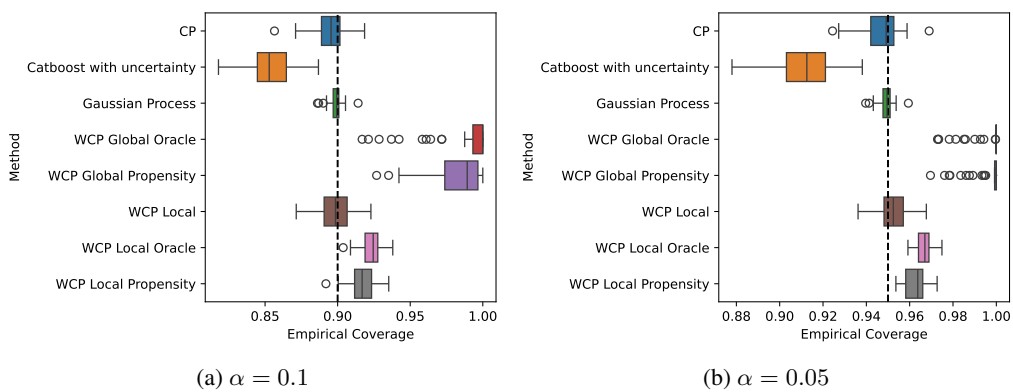

(a) $\alpha = 0.1$        (b) $\alpha = 0.05$

Figure 6: Barplot of the mean coverage calculated over 40 treatment values in 50 experiments for setup 2 scenario 2. Black dotted line is the ideal coverage.

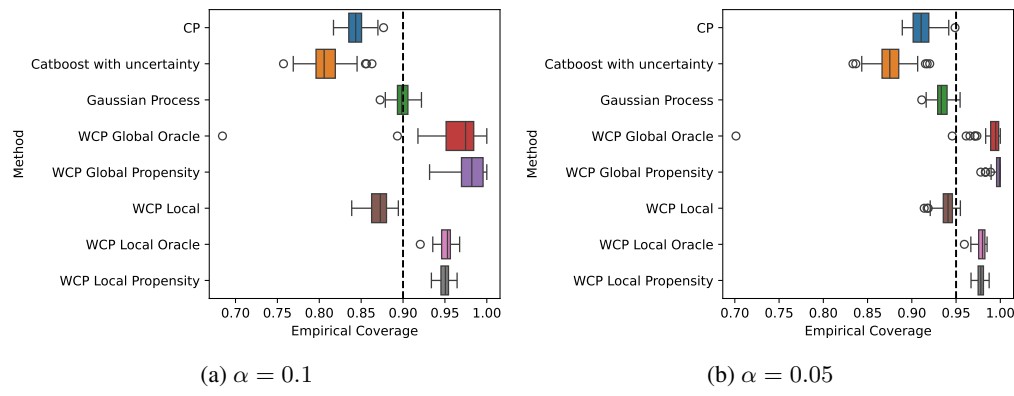

(a) $\alpha = 0.1$        (b) $\alpha = 0.05$

Figure 7: Barplot of the mean coverage calculated over 40 treatment values in 50 experiments for setup 1 scenario 1. Black dotted line is the ideal coverage.

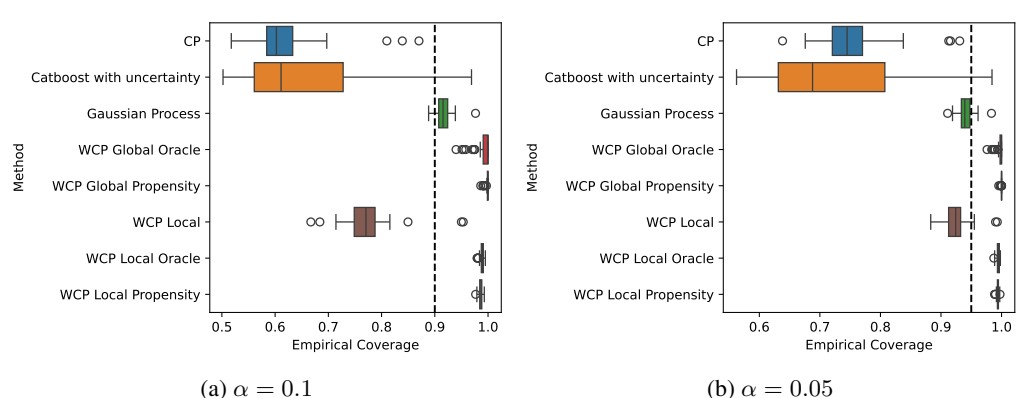

(a) $\alpha = 0.1$          (b) $\alpha = 0.05$

Figure 8: Barplot of the mean coverage calculated over 40 treatment values in 50 experiments for setup 1 scenario 2. Black dotted line is the ideal coverage.

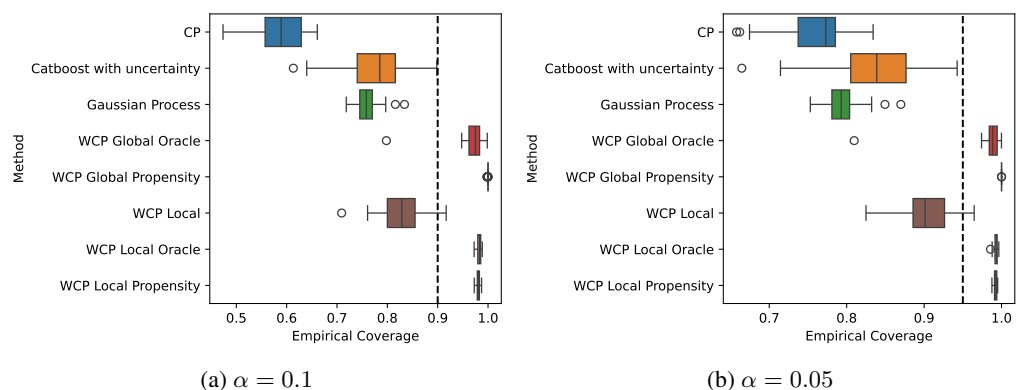

(a) $\alpha = 0.1$          (b) $\alpha = 0.05$

Figure 9: Barplot of the mean coverage calculated over 40 treatment values in 50 experiments for setup 1 scenario 3. Black dotted line is the ideal coverage.

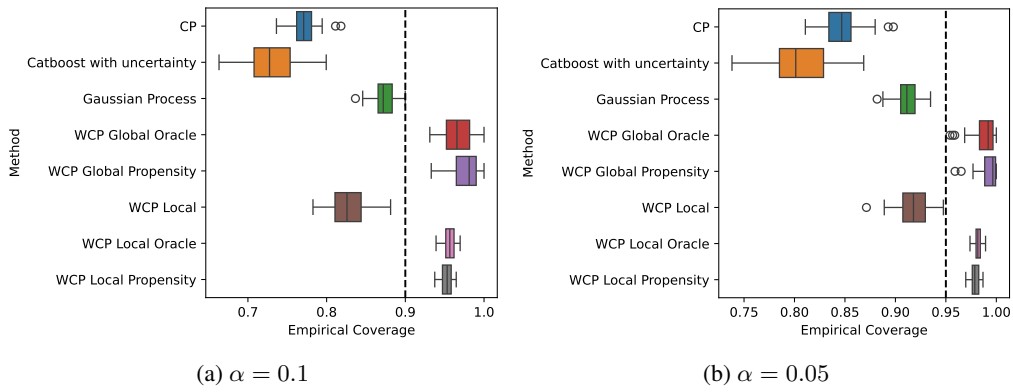

(a) $\alpha = 0.1$          (b) $\alpha = 0.05$

Figure 10: Barplot of the mean coverage calculated over 40 treatment values in 50 experiments for setup 1 scenario 4. Black dotted line is the ideal coverage.

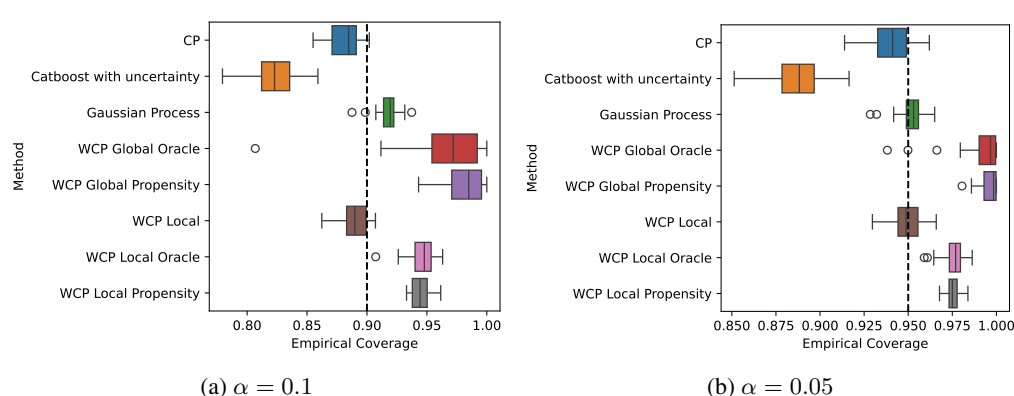

(a) $\alpha = 0.1$        (b) $\alpha = 0.05$

Figure 11: Barplot of the mean coverage calculated over 40 treatment values in 50 experiments for setup 1 scenario 5. Black dotted line is the ideal coverage.

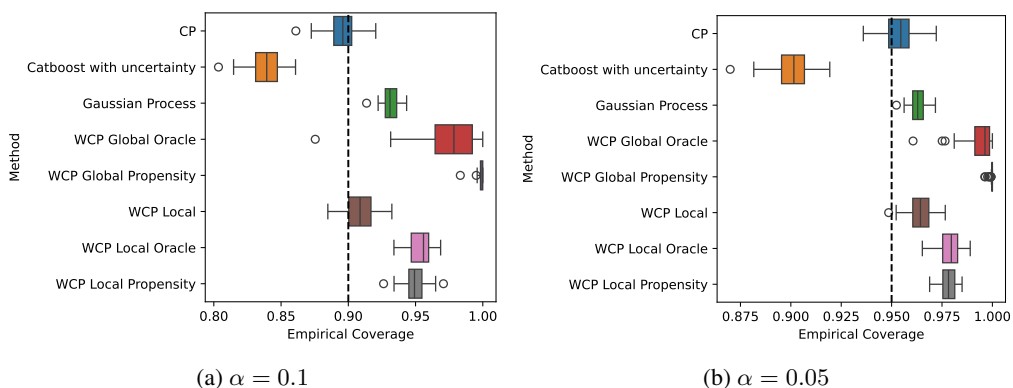

(a) $\alpha = 0.1$        (b) $\alpha = 0.05$

Figure 12: Barplot of the mean coverage calculated over 40 treatment values in 50 experiments for setup 1 scenario 6. Black dotted line is the ideal coverage.

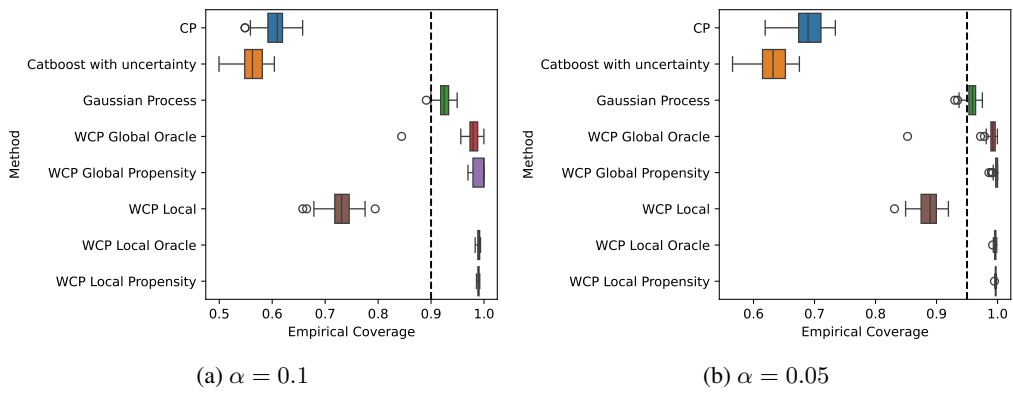

(a) $\alpha = 0.1$        (b) $\alpha = 0.05$

Figure 13: Barplot of the mean coverage calculated over 40 treatment values in 50 experiments for setup 1 scenario 7. Black dotted line is the ideal coverage.

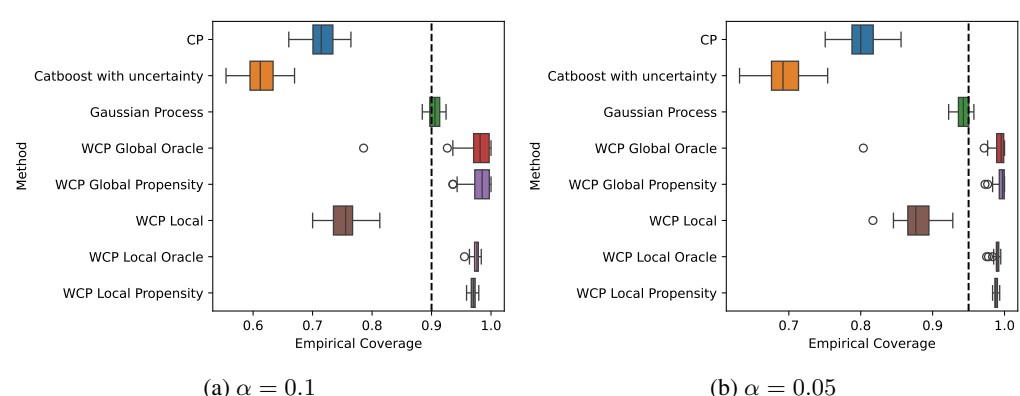

(a) $\alpha = 0.1$                    (b) $\alpha = 0.05$

Figure 14: Barplot of the mean coverage calculated over 40 treatment values in 50 experiments for setup 1 scenario 8. Black dotted line is the ideal coverage.

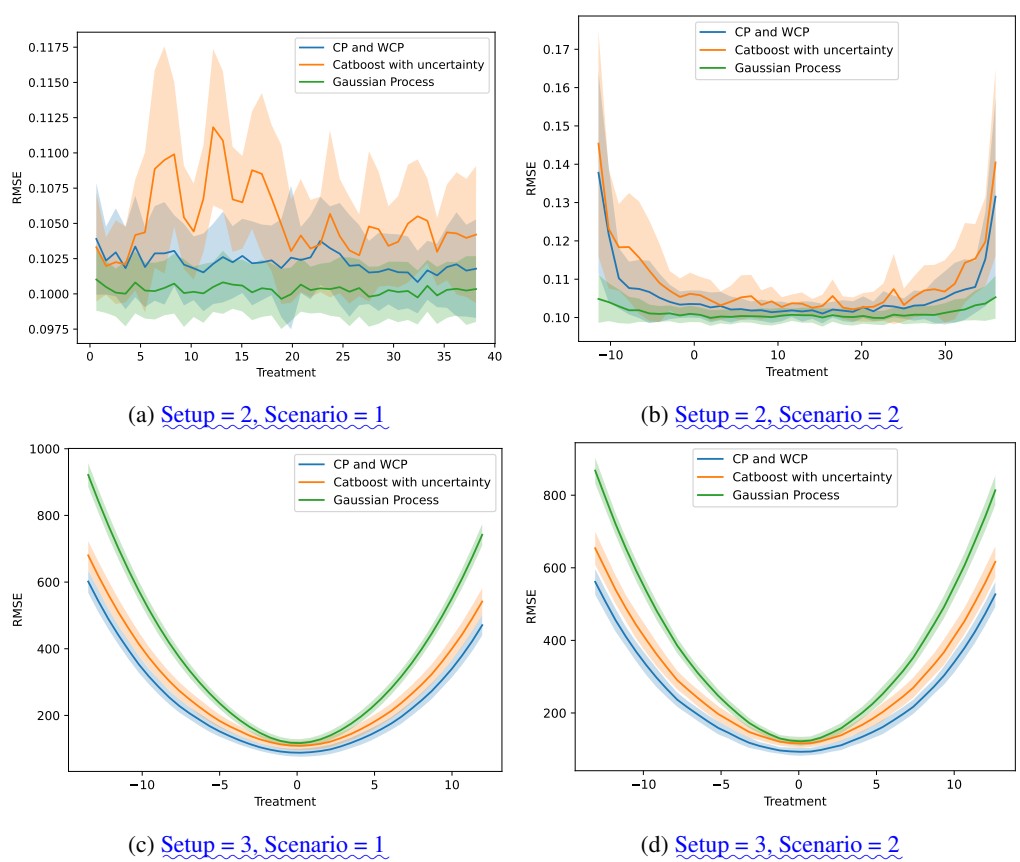

(a) Setup = 2, Scenario = 1          (b) Setup = 2, Scenario = 2

(c) Setup = 3, Scenario = 1          (d) Setup = 3, Scenario = 2

Figure 15: Plot of the CADRF RMSE with ± RMSE standard deviation across all repeated experiments for the considered treatment values for setup 2 and setup 3. As All WCP and CP methods use the same fitted base CatBoost CADRF learner they are represented by "CP and WCP".

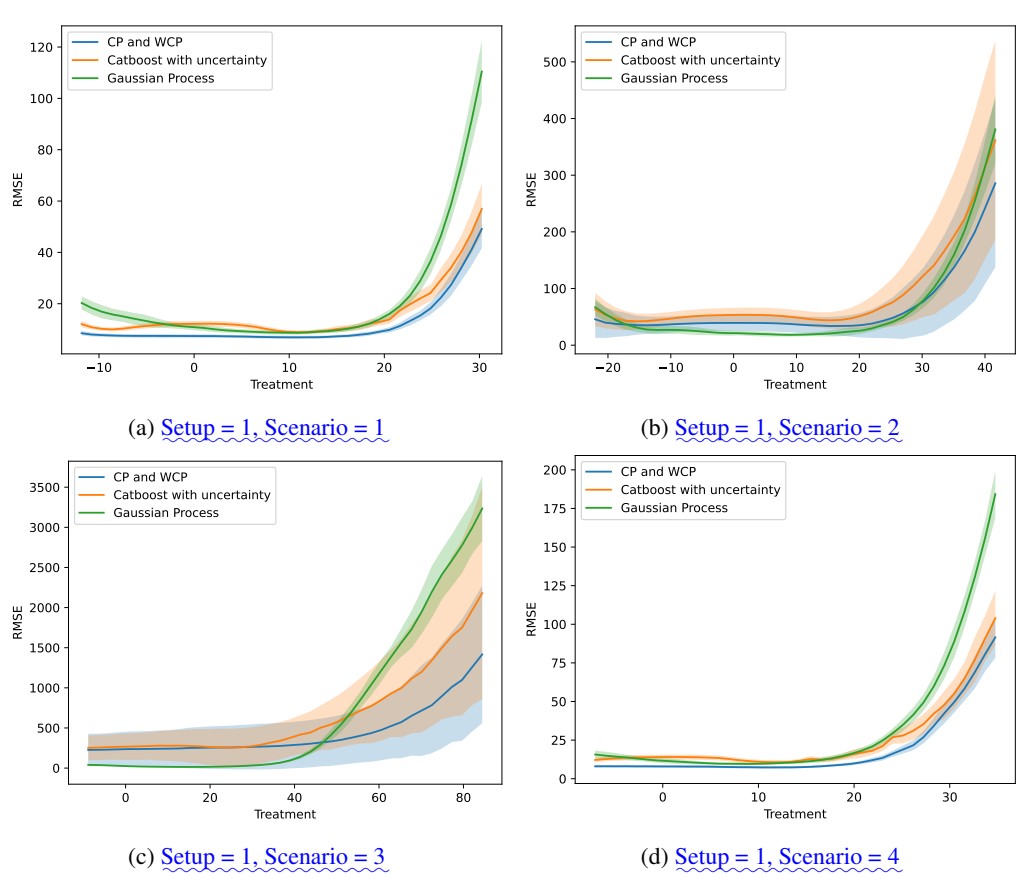

(a) Setup = 1, Scenario = 1

(b) Setup = 1, Scenario = 2

(c) Setup = 1, Scenario = 3

(d) Setup = 1, Scenario = 4

Figure 16: Plot of the CADRF RMSE with ± RMSE standard deviation across all repeated experiments for the considered treatment values for setup 1, scenarios 1 to 4. As All WCP and CP methods use the same fitted base CatBoost CADRF learner they are represented by "CP and WCP".

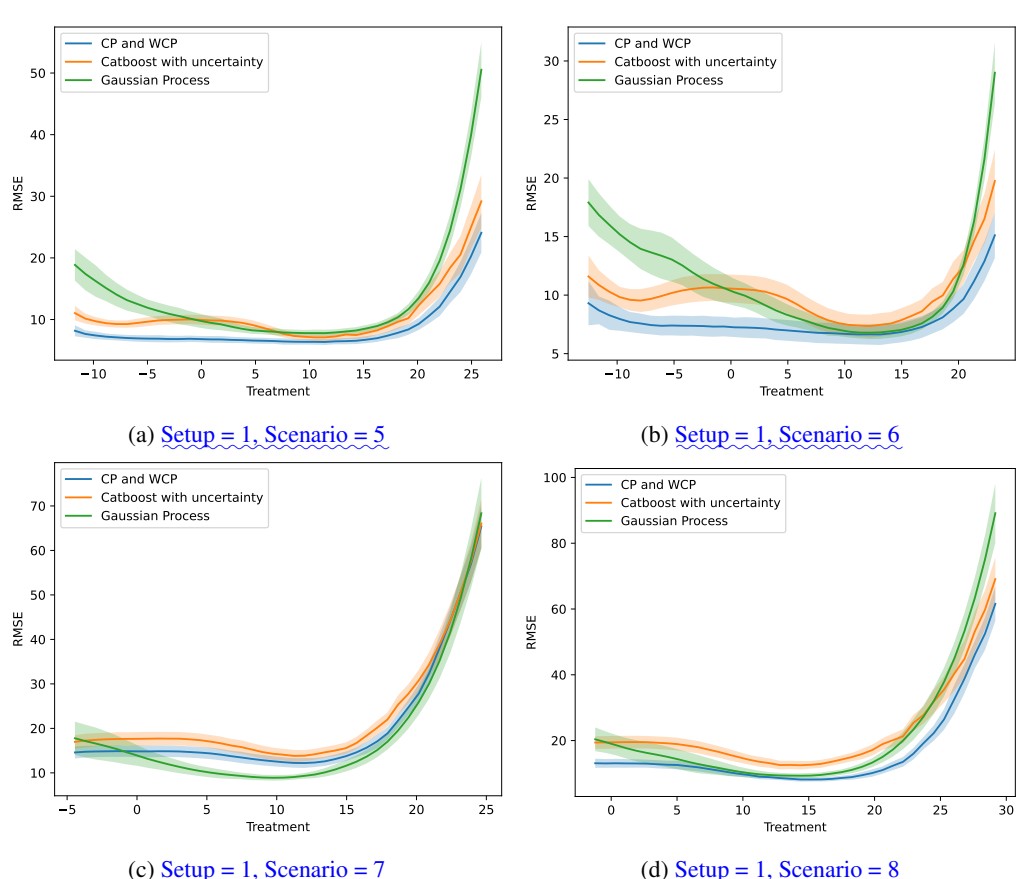

(a) Setup = 1, Scenario = 5

(b) Setup = 1, Scenario = 6

(c) Setup = 1, Scenario = 7

(d) Setup = 1, Scenario = 8

Figure 17: Plot of the CADRF RMSE with ± RMSE standard deviation across all repeated experiments for the considered treatment values for setup 1, scenarios 5 to 8. As All WCP and CP methods use the same fitted base CatBoost CADRF learner they are represented by "CP and WCP".

