# OpenReview forum: "Conformal Prediction for Dose-Response Models with Continuous Treatments"
_ICLR.cc/2025/Conference — Submitted to ICLR 2025_

### Official Review · Reviewer_z5PW · 2024-10-18

**Soundness:** 3
**Presentation:** 2
**Contribution:** 1
**Rating:** 5
**Confidence:** 3

**Summary:**

This paper addresses continuous treatment’s CATE via weighted conformal prediction.

**Strengths:**

This paper targets a significant and challenging task of considering uncertainty in CATE estimation when treatment is continuous.

**Weaknesses:**

-	Methodological contribution compared to prior work is incremental. The proposed idea of estimating counterfactual outcome interval using weighted conformal prediction has already been published by Lei et al. Lei et al. have proven that a generalized propensity score can be used for the weight in conformal prediction. The main difference between this paper and the similar work by Lei et al. is estimation targets (continuous CATE in this paper vs. discrete CATE in prior work).
-	Compared to another prior work by Schroder et al., this paper’s methodological contribution is also marginal. The discussion in Supplement C is not fully convincing in distinguishing this paper’s contribution from the prior work.
-	Novelty is limited. This paper applies an existing method to an existing task. No new approach or new generalizable insight was provided.
-	Validation is limited. Neither a formal theoretical guarantee nor empirical validation with real data were provided. I understand the lack of ground truth in the CATE world, but I would appreciate it if a theoretical guarantee could supplement the synthetic data validation. No comparison to baseline models.
-	Therefore, this paper does not have a broader impact on the following works in this field.


Reference

Lihua Lei, Emmanuel J. Candès, Conformal Inference of Counterfactuals and Individual Treatment Effects, Journal of the Royal Statistical Society Series B: Statistical Methodology, Volume 83, Issue 5, November 2021, Pages 911–938, https://doi.org/10.1111/rssb.12445

**Questions:**

Clarifying clear differences to prior similar works,
Convincing validation

---

> ### Author Response · Authors · 2024-11-18
> **Response to review**
>
> Thank you for your review. We addressed your questions and addressed weaknesses as follows.
>
> **Methodological Contribution & Novelty:** We acknowledge that the method builds on prior work, including the works of Tibshirani et al. (2019) and Lei and Candes (2021). However, our work focuses on the conditional average dose-response function (CADRF) compared to CATE estimation, which quantifies the causal effect, while we aim to provide dose-response curves within the potential outcome framework. Additionally, we generalize the work of Lei and Candes, which considers counterfactual inference in the binary treatment setting, to the continuous treatment setting. In the appendix (Appendix A, revised paper), we added the theoretical coverage guarantees of our proposed approach, together with a discussion of the desired coverage guarantees. We additionally show here that the same coverage guarantee as in Lei and Candes for binary treatment is impossible for continuous treatment without creating trivial intervals. Therefore, we introduce the context of a shift in the treatment distribution to the interventional distribution. For a more in-depth discussion see Appendix A (see revised paper).
>
> Additionally, we included recent work by Schroder et al., as it is closely related to our approach, though published contemporaneously (per ICLR 2025 guidelines). We included this for transparency and comparison. Additionally, to our knowledge, Schroder et al.'s current approach would be computationally infeasible in our simulation setting (time-wise), which reduces practical utility.
>
> **Theoretical Guarantee & Validation:** We have added a theoretical guarantee in Appendix A to support our results on synthetic datasets. In the Experiments section, we clarified that models like Gaussian Processes, Conformal Prediction, CatBoost with Uncertainty, and Local WCP serve as baseline models for comparison. These represent naive approaches to uncertainty quantification for dose-response curves. We are constrained by the lack of counterfactual observations to evaluate our method regarding real-world data. Hence, we added the theoretical guarantee.
>
> **Extensions & Applications:** We also expanded Appendix D to discuss potential applications and extensions, further highlighting the method's utility in real-world contexts, including clinical trials and other decision-making domains.

---

> > ### Comment · Reviewer_z5PW · 2024-11-26
> > **thanks for the response**
> >
> > Thanks for your response. I've read it and kept my score.

---

### Official Review · Reviewer_Jzrd · 2024-10-23

**Soundness:** 2
**Presentation:** 2
**Contribution:** 2
**Rating:** 8
**Confidence:** 4

**Summary:**

This paper introduces a novel methodology for conformal prediction (CP) in dose-response models with continuous treatments, aiming to provide uncertainty quantification (UQ) for individualized decision-making. The approach leverages propensity score estimation and weighted conformal predictive systems to generate prediction intervals across a continuous range of treatments, which is essential for personalized healthcare and other decision-critical fields. By incorporating covariate shift assumptions and using kernel-based weighting, the authors propose a robust solution for achieving local coverage of dose-response predictions. The paper is validated on synthetic datasets, demonstrating the effectiveness of the proposed method.

**Strengths:**

1. he paper presents an original application of conformal prediction to continuous treatment dose-response models, addressing an important gap in causal inference research. The integration of propensity score weighting and kernel-based adjustments to conformal prediction is a creative approach to ensure coverage under covariate shifts.
2. The paper is mostly clear, with well-structured sections that logically progress through the problem, related work, methodology, and experiments. The use of figures and visualizations to depict coverage is helpful for interpreting the results.
3. The problem of providing reliable prediction intervals for dose-response models has practical implications in many fields, such as personalized medicine, and this work represents a step forward in providing UQ in such contexts.

**Weaknesses:**

1. While the application is new, much of the methodology builds on existing CP and propensity score techniques without introducing fundamentally new theoretical contributions. The added value lies in the application context, but more could be done to differentiate this work from prior studies.
2. The reliance on synthetic datasets raises concerns about the method's practical utility. A more thorough evaluation on real-world data would strengthen the paper’s claim of addressing practical challenges in dose-response modeling.
3. Although the authors mention the efficiency improvements from weighted conformal prediction, the scalability of the method, particularly in real-time applications, remains unclear. Detailed analysis of the computational overhead, especially with large-scale data, would be beneficial.

**Questions:**

1. How does the method perform when applied to real-world dose-response data, particularly in scenarios where confounding factors are not as easily modeled as in synthetic datasets?
2. Can the proposed method scale to larger datasets with higher-dimensional covariates and continuous treatments without a significant increase in computational time?
3. How robust is the propensity estimation in cases where the true propensity distribution is unknown or difficult to estimate? What are the limitations when using kernel density estimation (KDE) in practice?
4. Beyond healthcare, what other domains have been considered for the application of this method, and how would the assumptions about covariate shift differ in these contexts?

---

> ### Author Response · Authors · 2024-11-18
> **Response to review**
>
> We would like to thank the reviewer for their thoughtful feedback and comments. We addressed the questions and weaknesses as follows.
>
> **Theoretical Contributions & Methodology:** While the methodology builds on existing CP and propensity score techniques, we believe the novel contribution lies in its application to dose-response modelling with uncertainty quantification for counterfactual treatments. We have included potential extensions and further applications of the method in Appendix D to highlight its broader applicability. Additionally, we present a more general theoretical framework for counterfactual inference and WCP in the revised paper (see Appendix A, revised paper).
>
> **Real-World Data & Practical Utility and Application Beyond Healthcare:** We agree that real-world validation is crucial. However, real-world evaluation is challenging due to the need for counterfactual observations of all possible treatments for each individual. In practice, real-world data typically only observes a single treatment per individual. Therefore, coverage guarantees cannot be measured without counterfactuals. However, in Appendix A we added a theoretical guarantee to support our coverage guarantee claims,  to compensate for this lack of real-world application. However, we discuss potential real-world applications in Appendix D, where the method can be applied to fields such as clinical trials, preventive maintenance, and sales. The interventional distribution assumptions about covariate shift remain similar across domains, as the goal is to consider all treatments equally in decision-making. If covariate shifts between train and test in the features $X$ are measurable or known, they can be incorporated into the method’s weights.
>
> **Scalability & Computational Overhead:** We have added a computational overhead analysis in Appendix C. The overhead scales linearly with the number of treatments evaluated for Local Propensity WCP (the most computationally intensive). Calibrations can be performed beforehand, and the primary complexity depends on the base learner. Treatment evaluations can be parallelized in real-time settings, allowing for inference in the second-to-millisecond range, depending on the base learner's inference time.
>
> **Robustness of Propensity Estimation:** We have included a robustness analysis in Appendix A, which discusses the relationship between errors in propensity estimates and the method’s coverage guarantees. The main limitation of KDE is the sensitivity to kernel choice and hyperparameter tuning which must be considered when implementing our version for propensity estimation. However, KDE is primarily used for smoothing and generating continuous density functions. Our Local Propensity WCP method is not limited to the CPS with KDE approach.

---

### Official Review · Reviewer_DXMN · 2024-11-03

**Soundness:** 1
**Presentation:** 2
**Contribution:** 1
**Rating:** 3
**Confidence:** 4

**Summary:**

In the manuscript, the authors propose a conformal prediction based method to obtain the interval estimation of the potential outcomes under continuous treatment.
To achieve this, the authors use the weighted conformal prediction method.
They also aim to provide a local guarantee for the proposed method via using the kernel weighting function.

**Strengths:**

1. The authors address a crucial issue in causal inference: estimating potential outcomes under continuous treatment. They also aim to provide a local guarantee for their proposed method, which is highly important in practical applications.

2. They have good literature review and make readers understand the background of the problem easily.

3. The method is relatively simple and easy to implement.

**Weaknesses:**

1. While the authors provide a method, they can not provide a theoretical guarantee for the proposed method. This is a significant drawback of the paper.

2. In my opinion, they do not illustrate the method well. The paper would benefit from more detailed illustrations for example an Algorithm or a flowchart.

3. The method relies on in my opinion a strong assumptions, that is interventional distribution is Uniform and there is not distributional shift between the training and test data in terms of $\mathbf{X}$.

4. The numerical experiments are not comprehensive enough and no real data application is provided.

**Questions:**

1. In the method, they mentioned they use Conformal Prediction System (CPS), however, I do not see it in the Method section. Only in the numerical experiments, they mention it.However, it is not clear how they use it.

2. The numerical experiments are confusing to me. They consider eight different methods for comparison, but it is not clear to me which methods are their proposed methods. What is the difference between these methods, such as WCP local and WCP global?


3. I think covariate shift is a very common issue in causal inference, why the authors assume there is no distributional shift between the training and test data in terms of $\mathbf{X}$?

4. Is the uniform distribution assumption for the interventional distribution realistic?

---

> ### Author Response · Authors · 2024-11-18
> **Response to review**
>
> Thank you for your review. Below, we address each of your concerns in detail.
>
> **Theoretical Guarantee:** We have added a theoretical finite-sample coverage guarantee, with a lower and upper bound, for any interventional distribution method using both oracle and estimated propensity function in Appendix A (see revised paper). This formalizes the method’s performance and provides the necessary theoretical foundation.
>
> **Illustrations and Algorithm:** To clarify the method, we have included pseudocode for both Global and Local Propensity WCP in Appendix C. This should help readers understand the fitting, calibration, and inference processes. We believe this addition also provides a clearer illustration of the method's workflow. This also clearly illustrates now that CPS is used to estimate the propensity function in combination with KDE, which was also already mentioned in Section 5.2; however, it was not that clear; hence we rephrased it (see line 384, revised paper).
>
> **Assumptions:** Regarding the assumption of no covariate shift between training and test data, we agree this is a simplifying assumption. Appendix D discusses potential extensions of both Global and Local Propensity WCP that account for covariate shifts in $X$. While this assumption was used to simplify the derivations, the method can handle covariate shifts in practice if we account for them in the weigths. If covariate shifts are measurable, they can be easily incorporated into both Global and Local Propensity WCP, and we have updated the methods section to include a reference to this.
> Regarding the uniform interventional distribution assumption, the uniform distribution comes from a decision-making standpoint when an intervention has yet to be performed, and we want to evaluate every treatment value equally. Hence we want an unbiased uncertainty quantification where we aim to evaluate all treatment values equally, as in a clinical trial. We added more nuance to the methodology section to clarify this.
>
> The coverage guarantee in Appendix A (see revised paper) is also general, allowing the use of any interventional distribution; our proposed interventional distributions are also discussed and translated into the general theoretical framework.
>
> **Numerical Experiments and Method Comparison:** We have updated the experiments section to clarify the differences between the various methods. Specifically:
> - **WCP Local Propensity** and **WCP Global Propensity** are our contributions, with **WCP Local Propensity** being the primary focus of this work.
> - **WCP Global** uses the global weights $w_{g,p}$, while **WCP Local** uses local weights $w_{l,p}$ as outlined in the methodology.
> - The other methods, such as CP, Gaussian Processes, CatBoost with Uncertainty, and Local WCP, serve as baseline comparisons.
>
> **Real-World Data and Application:** We agree that real-world application is important. However, evaluating the method with real-world data is challenging due to the inherent limitations of only observing a single treatment for each individual. Evaluating our method requires counterfactuals for all possible treatments for a single individual, which is not feasible with real-world data where only one treatment is observed. Thus, we focus on synthetic data to evaluate coverage guarantees. While we recognize this limitation, real-world validation is an area for future exploration; therefore, we added a discussion on the potential applications to Appendix D to cover this as well.

---

> > ### Comment · Reviewer_DXMN · 2024-11-25
> >
> > Thank you for the authors' rebuttal. I believe my concerns were not fully addressed, so I will maintain my original score.

---

> ### Author Response · Authors · 2024-11-25
>
> Dear Reviewer
>
> Thank you for the detailed review and the opportunity to respond. I’d like to briefly follow up to ensure our clarifications were fully understood. Specifically:
>
> 1. **Theoretical Guarantee:** We have added finite-sample coverage guarantees in Appendix A, addressing the lack of theoretical foundation mentioned in the review.
>
> 2. **Illustrations:** To improve clarity, pseudocode and additional details about CPS integration have been added in Appendix C and Section 5.2.
>
> 3. **Assumptions:** While our method assumes no covariate shift for simplicity, Appendix D now discusses extensions to handle shifts, making the method adaptable to practical scenarios. Similarly, the uniform interventional distribution is clarified as a design choice for unbiased evaluation, and any other distribution (even the dirac delta) can be used as illustrated in Appendix A.
>
> 4. **Clarification in Numerical Experiments:** The distinctions between our methods (WCP Local, WCP Global) and baselines are now explicitly clarified in the revised manuscript.
>
> We believe these changes address the concerns raised in the review. If there are remaining questions or ambiguities, we would be grateful for further guidance.
>
> Thank you for your time and consideration.

---

### Official Review · Reviewer_ptu1 · 2024-11-07

**Soundness:** 2
**Presentation:** 3
**Contribution:** 2
**Rating:** 5
**Confidence:** 2

**Summary:**

The paper introduces a new methodology for uncertainty quantification in dose-response models with continuous treatments using conformal prediction. ​ The approach leverages weighted conformal prediction, incorporating propensity estimation and kernel functions to address covariate shifts, ensuring coverage across all treatment values. ​ Building on the potential outcomes framework and generalized propensity scores, the method addresses some limitations in existing UQ techniques. ​ Experiments with synthetic data demonstrate its effectiveness, showing reliable prediction intervals with low treatment overlap. The practical implementation of this method can improve personalized dosing and interventions in various fields, enhancing decision-making by providing robust uncertainty quantification. ​

**Strengths:**

The paper introduces a novel approach using conformal prediction to uncertainty quantification in dose-response models. The use of weighted conformal prediction ensures coverage across all treatment values, even under covariate shifts. The methodology has practical implications for personalized healthcare, drug dosing, and other fields requiring individualized treatment decisions. ​

**Weaknesses:**

1. The accuracy of the method relies heavily on the quality of the propensity score estimation, which can be challenging in real-world scenarios. In Section 5.2, the paper discussed using both oracle and estimated propensity distributions. How robust are their results to potential errors or biases in propensity score estimation? A sensitivity analysis could provide insights into how variations in the quality of propensity score estimation impact the overall accuracy of their method.

2. The experiments are conducted on synthetic data, and the method's performance in real-world applications remains to be fully validated. ​

**Questions:**

How does the method perform with real-world data? It will make the method become more impactful and convincing with real data application analysis. I understand that applying real data for treatment effect estimation can be challenging, especially for continuous dose scenario. However, I encourage the authors to suggest specific real-world applications related to optimal dose recommendation, as this is an area where their method could provide significant insights. Probably some real data application deal with optimal dose level recommendation and use offline reward/value/outcome function to evaluate the performance of the estimated decision rule?

---

> ### Author Response · Authors · 2024-11-18
> **Response to review**
>
> We would like to thank the reviewer for their thoughtful feedback and comments. We addressed the questions and weaknesses as follows.
>
> We agree that real-world validation is crucial for demonstrating the practical applicability of our method. However, evaluating the model on real-world data would not verify the necessary coverage guarantees, as this would require counterfactual outcomes for every possible treatment value and every sample, which is typically unavailable in real-world settings. In practice, we only observe a single treatment per sample, while our method is designed to quantify uncertainty across all counterfactuals. Therefore, we evaluate our method using synthetic data, where the true counterfactuals are known, to ensure that the method works as expected.
>
> To address the need for coverage guarantees, we have included theoretical coverage guarantees in Appendix A (see revised paper) that formalize the coverage for all counterfactual treatments and give a lower and upper bound for the coverage when using both the oracle and estimated generalized propensity function. While synthetic data is helpful for this purpose, we recognize that real-world applications are the ultimate goal. To that end, we have expanded the appendix with a discussion of potential applications for our method, including clinical trials, preventive maintenance, and sales.
>
> One challenge in applying the method to real-world data is the lack of overlap in treatment distributions, especially in areas like drug dosing, where doses are determined by predefined treatment protocols, limiting the range of treatments across samples. Our method’s uncertainty quantification is intended to assist decision-making by accounting for counterfactuals. However, translating this uncertainty into actionable, optimal decisions remains an area for future work, particularly when uncertainty quantification could provide complete distributions, such as using CPS. For that domain, clinical utility, decision functions, or offline reward/value/outcome can be explored in future work.

---

### Official Review · Reviewer_Uate · 2024-11-09

**Soundness:** 3
**Presentation:** 4
**Contribution:** 3
**Rating:** 8
**Confidence:** 4

**Summary:**

The authors propose a conformal prediction-based method for estimating uncertainty in the dose-response function, which defines the effect of continuous treatment on a continuous outcome, in the presence of confounders. Their method uses weighted conformal prediction, with weights based on generalized propensity scores. The presentation is exemplary and instructive throughout, including the motivation for and description of the proposed method. Experiments cover two established simulation settings and one new simulation setting. Results show that the resulting prediction intervals tend to be conservative, in the sense that empirical coverage of the true dose-response function is higher than intended.

**Strengths:**

- Reliable dose-response estimation from observational data is important in medicine and other settings.
- The writing style, mathematical notation and presentation, and explanations of concepts are outstanding throughout.
- The methodology is novel to my knowledge and builds on recent progress in conformal prediction and causal methods for continuous treatments.
- Experimental settings and baseline methods are appropriate.
- The proposed method consistently achieves better empirical coverage than comparator methods.

**Weaknesses:**

- The evaluation is somewhat limited and focused almost entirely on empirical coverage.
- Error of the estimated CADRF is not presented except indirectly in Figure 2 for only one of the settings (Setup 3, Scenario 1).
- Empirical coverage is higher than desired in most cases and often very close to 1, and the prediction intervals are only shown for a single example.
- All this taken together makes me suspect that the method often yields excessively wide prediction intervals that may not be useful.
- The authors discuss the fact that the method yields conservative prediction intervals and provide brief explanations, but I think more discussion should be devoted to this given its central importance.
- I also think it is critical to provide figures akin to Figure 2 for more of the settings and compare error of the estimated CADRF between methods.

**Questions:**

My questions are implied by the weaknesses listed above. I'd like to see:
- more figures akin to Figure 2
- a comparison of error of the estimated CADRF between methods
- more commentary on why the method yields such conservative prediction intervals

Additionally:
- What are the implications of the conservative prediction intervals on usefulness of the method in practical settings?
- How might the method be improved subsequently to achieve ideal empirical coverage?

---

> ### Author Response · Authors · 2024-11-18
> **Response to review**
>
> We would like to thank the reviewer for their thoughtful feedback and comments. We addressed the questions and weaknesses as follows.
>
> **Error of Estimated CADRF:**
> We acknowledge that the error of the estimated CADRF was only shown indirectly in Figure 2 for one of the experimental setups. In response, we have added RMSE results for all experiments and treatment values in Appendix F. However, as our method is model-agnostic, the RMSE could be further improved by using more suitable models and tuning hyperparameters.
>
> **Conservative Intervals:**
> The conservativeness is entirely determined by the number of samples in the calibration set (in the case of split-WCP) and how well-behaved the likelihood ratio is; we included a more theoretical discussion of the upper bound of the coverage in Appendix A (see revised paper). If the prediction intervals become infinite, this indicates that in these regions, there is not enough data support (i.e., lack of overlap) for this sample to provide a counterfactual prediction. Thus, the model cannot be trusted here. For example, assume that the overlap or positivity assumption is violated, i.e., $\frac{d\tilde{P}{T|X}}{dP_{T|X}} = \infty$ in terms of the interventional distribution, this will result in the trivial interval $(-\infty, \infty)$, since $w(X_i)=0, \forall i \in [1,...,n]$ and $w(X_{n+1})=\infty$ resulting in $p^w_i(X_{n+1})=0, \forall i \in [1,...,n]$ and $p^w_{n+1} = 1$.
>
> The reason for quite conservative coverage in the experiments is that the covariate distribution $P_X$ remains fixed while we shift the treatment distribution. This can result in sharp likelihood ratios, reducing the effective sample rate. However, in places with enough overlap, the empirical coverage is close to the target coverage, which aligns with the theoretical results in Appendix A.
>
> To achieve empirical coverage closer to the target coverage, one could increase the calibration samples; another approach would be to use a smoothing term drawn from the uniform distribution, allowing exact coverage guarantees under the Oracle propensity function. However, this would result in non-deterministic prediction intervals.

---

> > ### Comment · Reviewer_Uate · 2024-11-18
> > **Response to rebuttal**
> >
> > Thanks for your response. I will be keeping my score (8: accept).

---

### Meta-Review · Area_Chair_HHWa · 2024-12-21

**Metareview:**

This paper proposed a conformal prediction-based method for uncertainty quantification in dose-response models with continuous treatments. The approach leverages weighted conformal prediction, incorporating propensity estimation and kernel functions to address covariate shifts. The problem is well motivated and the extension of conformal inference to continuous treatments is important. The initial reviews mostly questioned on the novelty and contribution of this work. Neither a formal theoretical guarantee nor empirical validation with real data were provided. While a great effort has been made by authors during rebuttal to address these issues, the main concerns remain. The proposed method is a direct extension or integration of conformal prediction and propensity score techniques without introducing fundamentally new theoretical contributions. Both the theoretical and methodological innovations are limited.

In any case, this is clearly a borderline paper. It is interesting but also has a low originality and weak significance. For that reason I think it is not ready for ICLR. We'd encourage the authors to take into consideration all the feedback provided by the reviewers to strengthen their manuscript for resubmission.

**Additional Comments On Reviewer Discussion:**

The initial reviews mostly questioned on the novelty and contribution of this work. Neither a formal theoretical guarantee nor empirical validation with real data were provided. While a great effort has been made by authors during rebuttal to address these issues, the main concerns remain that both the theoretical and methodological innovations are limited. After discussion with the reviewers, we agreed it is not quite ready for publication.

---

### Decision · Program_Chairs · 2025-01-22

Reject